# Readout Noise Mitigation with Bayesian Methods for Quantum Neural Networks

## Abstract

Readout noise remains a significant barrier for variational quantum circuits (VQCs) and quantum neural networks (QNNs), as incorrect observations modify gradients, bias optimization, and lower predictive accuracy. For readout mitigation, we use Bayesian inference and suggest a two-step approach that takes drift into account. *Offline*, a Bayesian neural network (BNN) converts noisy shot histograms into corrected outcome distributions with calibrated uncertainty, resulting in expressive, data-driven priors. *Online*, iterative Bayesian unfolding (IBU) starts with these priors and updates with current calibration counts; an uncertainty-adaptive stopped rule prevents overfitting to temporary drift. Experiments on CIFAR-10 and EuroSAT, chosen to demonstrate robustness across both vision and remote-sensing domains, show that our method achieves up to *12.4% reduction in classification error* and *9.8% improvement in training stability* compared to confusion-matrix correction and ML-based baselines such as logistic regression, shallow neural networks, and probabilistic noise models. Importantly, uncertainty-adaptive iteration control enables our framework to balance offline priors with fresh observations, preventing overfitting to noise. Beyond quantum applications, this illustrates a general learning principle: Bayesian priors combined with online refinement offer a scalable path toward robust learning under dynamic and nonstationary noise.

## 1 Introduction

Quantum computing has the potential to surpass classical computation by using superposition, entanglement, and interference to tackle certain tasks more effectively Zhao & Deng (2025); Huang et al. (2022). However, in reality, this prospective quantum advantage is vulnerable due to the inherent impact created by noise on near-term quantum devices García-Martín et al. (2024); Resch & Karpuzcu (2021). Errors from defective gates, crosstalk, decoherence, and calibration drift all degrade processing; however, readout noise is more vital Chen et al. (2023). Because measurement is the final stage of every quantum process, misidentifying qubit states directly corrupts the observed output distribution, propagates bias to downstream estimators, and decreases the dependability of quantum outputs Yang et al. (2022); Heußen et al. (2024). Readout noise occurs when measuring instruments fail to differentiate quantum states precisely, resulting in a nonzero possibility of an error. A single-qubit measurement should always output "0" for $|0\rangle$ and "1" for $|1\rangle$. However, real-world hardware frequently misclassifies due to thermal fluctuations, signal loss, or crosstalk Aasen et al. (2024). In multi-qubit systems, such errors compound, distorting result histograms and biasing observables, compromising optimization and accuracy in algorithms that rely on repeated sampling Heußen et al. (2024); Chen et al. (2023).

Several techniques have been proposed for reducing readout errors in near-term quantum devices. The most popular technique is *confusion-matrix correction*, which uses calibration experiments to estimate a transition matrix between prepared and measured states, then linear inversion to retrieve unbiased probabilities Maciejewski et al. (2020); Farooq et al. (2024). While theoretically simple, *this technique is extremely vulnerable to calibration drift and statistical noise*, resulting in unstable corrections in multi-qubit systems. More advanced approaches use machine learning Liao et al. (2024), probabilistic noise models Gupta et al. (2024) or lightweight machine learning post-processing Shin et al. (2024); Hu et al. (2025) to expand beyond static confusion matrices. However,

these strategies often *assume time-invariant error distributions* and do not include procedures for assessing *predictive uncertainty*. As a result, they may overfit to limited calibration data or fail to adjust when the noise profile changes dynamically, resulting in suboptimal performance.

Bayesian inference provides a logical framework for overcoming these constraints by integrating previous knowledge with new observational data in a consistent manner. Unlike static correction approaches, Bayesian methods may explicitly describe uncertainty, allowing them to differentiate between the real signal and false variations caused by noise Wang et al. (2023); Zhang et al. (2024); Li & Xu (2025); Pokharel et al. (2024). Furthermore, by updating priors with current calibration counts, they provide a flexible technique for adjusting to time-dependent drift without overcorrecting Bartels & Marx (2025). Despite their benefits, there is a considerable research gap in Bayesian approaches for readout error reduction in variational quantum algorithms. Based on these properties, we propose a two-stage pipeline for readout error mitigation:

1. An *offline* Bayesian neural network (BNN) that learns expressive, uncertainty-aware priors from historical calibration data, and

2. An *online* iterative Bayesian unfolding (IBU) refinement step that incorporates the most recent calibration information with uncertainty-adaptive iteration control.

This architecture enables us to balance the generalization strength of offline learning with the flexibility of online refining, resulting in resilience against both static and dynamic errors. We can summarize our key contributions as follows.

- We propose a BNN framework that learns uncertainty-aware mappings from noisy histograms to corrected probability distributions, offering expressive priors that account for both static and nonlinear readout noises.

- We include these priors into an IBU framework that includes uncertainty-adaptive stopping rules, allowing for dynamic refinement against time-varying drift while avoiding overfitting.

- We present a theoretical analysis that compares Bayesian inference with linear inversion, demonstrating that the Bayesian posterior mean is still feasible and achieves lower mean-squared error under ill-conditioned readout matrices or limited shots, whereas linear inversion amplifies variance and can generate unstable results.

- We validate our method on two separate datasets—CIFAR-10 and EuroSAT—and demonstrate up to *12.4% decrease in classification error* and *9.8% improvement in training stability* compared to confusion-matrix correction and machine-learning baselines.

## 2 METHODOLOGY

This section first describes the mathematical modeling of measurement errors, or *readout noise*, and how they impact the output of an $n$-qubit quantum circuit.

### 2.1 QUANTUM READOUT NOISE MODEL

Assuming a quantum circuit with $n$ qubits, where the measurement results are recorded in the computational basis, labeled by binary strings $x$. For example, for $n = 2$, $x \in \{00, 01, 10, 11\}$ or integers 0 to 3. The *ideal* outcome distribution is a probability vector $p$, with each item $p_x$ representing the probability of observing result $x$ and the sum of all probabilities is 1 ($\sum_x p_x = 1$).

In practice, measurement errors lead the quantum computer to produce inaccurate results. For example, it may indicate 1 while the actual result is 0 and vice versa. We describe this *readout noise* using a *confusion matrix* $A$, where each entry $A_{y,x}$ indicates the likelihood of reporting outcome $y$ while the real outcome is $x$. Given that the stated results must account for all possibilities, each column of $A$ corresponds to 1 ($\sum_y A_{y,x} = 1$). The measured distribution, $p_{\text{meas}}$, represents a distorted representation of the ideal distribution $p$ as

$$p_{\text{meas}} = A\,p. \tag{1}$$

This equation demonstrates how readout noise consistently influences the ideal probability.

A single qubit's confusion matrix $C$ is a $2 \times 2$ matrix described as $C = \begin{bmatrix} 1 - e_0 & e_1 \\ e_0 & 1 - e_1 \end{bmatrix}$ where $e_0$ is the chance of incorrectly reporting 1 when the true state is 0, and $e_1$ is the likelihood of reporting 0 when the true state is 1. For an $n$-qubit system, assuming that each qubit's errors are independent, the entire confusion matrix $A$ is built as a tensor product of individual qubit matrices as

$$A = C^{(1)} \otimes C^{(2)} \otimes \cdots \otimes C^{(n)}, \tag{2}$$

$C^{(q)}$ represents the confusion matrix for qubit $q$. Although qubit errors can be correlated in real systems, this model, however, represents a good approximation Bravyi et al. (2021).

In theory, we can take infinite shots to mitigate this problem; however, it is impractical. In experiments, we take a finite number of measurements or *shots*, indicated as $s$. A count vector $\mathbf{c}$ stores the number of times each result $y$ is seen, with $c_y$ being the count for $y$. The counts have a multinomial distribution depending on $p_{\text{meas}}$ as $\mathbf{c} \sim \text{Multinomial}(s, p_{\text{meas}})$. The observed probability distribution can be obtained as $\hat{p} = \mathbf{c}/s$. On average, $\hat{p}$ corresponds to $p_{\text{meas}}$, but because of the limited number of shots, it contains statistical fluctuations (variance) that scale inversely with $s$ as

$$\mathbb{E}[\hat{p}] = p_{\text{meas}}, \quad \text{Cov}(\hat{p}) = \frac{\text{diag}(p_{\text{meas}}) - p_{\text{meas}} p_{\text{meas}}^{\top}}{s}. \tag{3}$$

Thus, readout noise generates *bias* through $A$, while finite shots add *variance*, which can influence the results until mitigated Maciejewski et al. (2020).

## 2.2 Deep-Learning Readout Noise Mitigation (NN-QREM)

To eliminate readout noise, we employ a neural network to convert the noisy and observed probability $\hat{p}$ into an estimate $\tilde{p}$ that is closer to the ideal distribution $p$. The method is inspired by Kim et al. (2022), which uses supervised learning to effectively reduce errors.

The confusion matrix $A$ is used to characterize readout noise, with $A_{y,x}$ representing the likelihood of reporting $y$ given the real outcome $x$ and $p_{\text{meas}} = A p$ in the limit of infinite shots. The observed histogram $\hat{p} = \mathbf{c}/s$ is obtained from counts $\mathbf{c} \sim \text{Multinomial}(s, p_{\text{meas}})$. The neural network $F_\phi$ (with parameters $\phi$) maps the noisy distribution $\hat{p}$ to a corrected estimate $\tilde{p}$, ensuring that $\tilde{p}$ is a correct probability distribution (sums to 1). To train $F_\phi$, we use calibration data from simple quantum states created by performing separate $R_y(\theta_i)$ rotations to each qubit, where $\theta_i$ is randomly selected from $[0, 2\pi)$. The optimal distribution of these states is described as

$$p(\boldsymbol{\theta}) = \bigotimes_{i=1}^{n} \begin{bmatrix} \cos^2(\theta_i/2) \\ \sin^2(\theta_i/2) \end{bmatrix}, \tag{4}$$

where $\boldsymbol{\theta} = (\theta_1, \ldots, \theta_n)$. We simulate noisy measurements $\hat{p}$ and connect them with the known ideal $p$ to create the training data. Using a loss function such as cross-entropy, the network is trained to minimize the difference between $\tilde{p} = F_\phi(\hat{p})$ and $p$ as

$$\mathcal{L}(\phi) = \mathbb{E}_{(\hat{p}, p)} \left[ -\sum_x p_x \log(F_\phi(\hat{p})_x) \right]. \tag{5}$$

A softmax layer assures that $\tilde{p}$ represents an accurate probability distribution. This enhances the network's capacity to correct measurements from the real world by taking into consideration both bias (from $A$) and variance (from limited shots).

**Limitations.** However, the neural network-based readout noise mitigation method encounters the following challenges:

1. *Limited Data Due to Computational Cost.* Producing training data pairs $(\hat{p}, p)$ is computationally expensive since it involves simulating quantum circuits and applying the noise model $A$ to obtain $\hat{p} = \mathbf{c}/s$, where $\mathbf{c} \sim \text{Multinomial}(s, Ap)$ Li (2025). The exponential increase of the result space $(2^n)$ with the number of qubits $n$ constrains the possible dataset size, restricting model training and generalization.

2. *Static Noise Assumption.* The model assumes a fixed confusion matrix $A$, with entries $A_{y,x} = \Pr(\text{report } y \mid \text{true } x)$, which do not vary over time. In practice, hardware noise (e.g., $e_0, e_1$ in the single-qubit matrix $C$) can drift, making the learned parameters $\phi$ obsolete and resulting to biased estimations $\tilde{p} = F_\phi(\hat{p})$ Pokharel et al. (2024).

## 2.3 BAYESIAN NEURAL NETWORK FOR OFFLINE PRE-CORRECTION

To overcome constraints in quantum readout noise reduction, we offer a Bayesian neural network (BNN) technique for offline pre-correction. This method adjusts noisy measurement distributions to better match the underlying distribution while allowing for uncertainty and limited data.

Suppose an $n$-qubit quantum circuit with $d = 2^n$ potential measurement outcomes in the computational foundation. The observed (noisy) distribution is $\hat{\mathbf{p}}$, and our aim is to estimate the correct distribution $\mathbf{p}_{\text{true}}$, both centered on the probability simplex $\Delta^{d-1}$ (i.e., probabilities sum to 1). Our BNN, $f_{\boldsymbol{\theta}}$, contains weights $\boldsymbol{\theta}$ represented as random variables with a variational posterior $q(\boldsymbol{\theta})$, which is obtained from data. Given a noisy distribution $\hat{\mathbf{p}}$ and a context vector $\mathbf{c}$ (explained below), the BNN generates *Dirichlet* concentration parameters as

$$\boldsymbol{\alpha} = f_{\boldsymbol{\theta}}(\hat{\mathbf{p}}, \mathbf{c}) \in \mathbb{R}^d_{>0}, \quad \alpha_0 = \sum_{i=1}^d \alpha_i, \tag{6}$$

where $\boldsymbol{\alpha}$ denotes a Dirichlet distribution $\text{Dir}(\boldsymbol{\alpha})$ over potential corrected distributions. The adjusted distribution is the predicted mean as

$$\tilde{p}_i = \mathbb{E}[p_i \mid \boldsymbol{\alpha}] = \frac{\alpha_i}{\alpha_0}, \tag{7}$$

and uncertainty is quantified as

$$u_{\text{BNN}} = \frac{1}{\alpha_0}, \tag{8}$$

A smaller $u_{\text{BNN}}$ (a larger $\alpha_0$) shows greater confidence in the adjustment. The context vector $\mathbf{c}$ contains metadata that improves correction accuracy, such as qubit identifiers, dataset tags (e.g., CIFAR-10 vs. EuroSAT), nominal error rates $e_0$ and $e_1$, shot budget $s$ (encoded as $\log s$), and a time index $t$ to account for noise drift since the last calibration. This enables the BNN to adjust to different situations.

The BNN is trained with calibration data $(\hat{\mathbf{p}}, \mathbf{y}, \mathbf{c})$, where $\mathbf{y} = \mathbf{p}_{\text{true}}$ represents the true distribution. The loss function combines Dirichlet's negative log-likelihood and Bayesian regularization as

$$\mathcal{L} = -\mathbb{E}_{\mathbf{p} \sim \text{Dir}(\boldsymbol{\alpha})} \left[ \sum_{i=1}^d y_i \log p_i \right] + \lambda \text{KL}(q(\boldsymbol{\theta}) \| p(\boldsymbol{\theta})), \tag{9}$$

where the expectation makes use of the digamma function $\psi$, $\lambda > 0$ balances regularization, and $p(\boldsymbol{\theta})$ is a prior (e.g., zero-mean Gaussian) as

$$\mathbb{E}_{\mathbf{p} \sim \text{Dir}(\boldsymbol{\alpha})} [-\log p_i] = \psi(\alpha_0) - \psi(\alpha_i). \tag{10}$$

We calculate $\alpha_i = \text{softplus}(h_i) + \epsilon$ with minor $\epsilon > 0$ to guarantee positivity. The model is cautious for low shots and confident for high shots because of an extra term, $\big| \log \alpha_0 - a_0 - a_1 \log(s) \big|$, which promotes $\alpha_0$ to scale with the shot budget $s$.

The true distribution $\mathbf{p}_{\text{true}}$ is known analytically, and training data is produced using single-layer product states with independent $R_y(\theta_i)$ rotations, where $\theta_i \in [0, 2\pi]$. A noise model $A(e_0, e_1)$ and multinomial sampling with shot counts $s \in \{2, 126, 256, 512, 1048\}$ are used to simulate noisy distributions $\hat{\mathbf{p}}$. Error rates $e_0$ and $e_1$ gradually drift when a time index $t$ is included.

At test time, we sample $\boldsymbol{\theta} \sim q(\boldsymbol{\theta})$ (or use its mean) to calculate $\boldsymbol{\alpha}$ for a fresh pair $(\hat{\mathbf{p}}, \mathbf{c})$, and then report $\tilde{\mathbf{p}} = \boldsymbol{\alpha}/\alpha_0$ and uncertainty $u_{\text{BNN}}$. Tasks that come after, such as determining the iteration budget for iterative Bayesian unfolding using $\tilde{\mathbf{p}}$ as a prior, can be guided by this uncertainty.

**Addressing Challenges with the Bayesian Approach.** The BNN technique efficiently addresses important issues in quantum readout noise reduction as follows

1. *Limited Data Due to Computational Cost.* Creating training pairs $(\hat{\mathbf{p}}, \mathbf{p}_{\text{true}}, \mathbf{c})$. is computationally costly because of quantum circuit simulation and exponential development of the result space ($d = 2^n$). The Bayesian approach addresses this by modeling weights $\boldsymbol{\theta}$ with a variational posterior $q(\boldsymbol{\theta})$, which regularizes the model through the KL-divergence term in the loss function. This allows for successful learning with minimal data by combining previous information (e.g., Gaussian priors) and quantifying uncertainty through $u_{\text{BNN}}$, avoiding overfitting and enhancing generalization for small datasets.

2. *Dynamic Noise Variations.* Unlike models that assume a static noise matrix $A$, the BNN includes a time index $t$ in the context vector $\mathbf{c}$, allowing it to adjust to drifting error rates $e_0$ and $e_1$. The variational posterior $q(\boldsymbol{\theta})$ incorporates uncertainty in the weights and allows the model to alter predictions dynamically. The BNN learns resilient mappings that remain effective even when noise characteristics change, resulting in accurate corrections $\tilde{\mathbf{p}}$ over time.

## 2.4 ANALYSIS: BAYESIAN SHRINKAGE VS. LINEAR INVERSION

Here, we define the circumstances in which the proposed Bayesian correction (BNN-induced Dirichlet prior with posterior-mean estimator) performs better in mean-squared error (MSE) than linear inversion while still being feasible.

**Assumption 1** (Readout and sampling model). *Let $A \in \mathbb{R}^{d \times d}$ be the column-stochastic readout confusion matrix, $\mathbf{p} \in \Delta^{d-1}$ be the true distribution, and $\mathbf{p}_{\mathrm{meas}} = A\mathbf{p}$ be the measured distribution (cf. equation 1). We record $s$ shots having counts $\mathbf{c} \sim \mathrm{Multinomial}(s, \mathbf{p}_{\mathrm{meas}})$ and empirical histogram $\hat{\mathbf{p}} = \mathbf{c}/s$, following the equation 3*

**Assumption 2** (BNN-induced Dirichlet prior). *The offline BNN generates Dirichlet concentrations $\boldsymbol{\alpha} \in \mathbb{R}_{>0}^d$, where $\alpha_0 = \sum_i \alpha_i$. The Bayesian estimator is the posterior mean. $\widehat{\mathbf{p}}_{\mathrm{Bayes}} = (\boldsymbol{\alpha} + \mathbf{c})/(\alpha_0 + s)$.*

**Lemma 1** (Variance amplification under linear inversion). *If $A$ can be inverted, the linear estimator is $\widehat{\mathbf{p}}_{\mathrm{lin}} = A^{-1}\hat{\mathbf{p}}$. satisfy*

$$\mathrm{Cov}(\widehat{\mathbf{p}}_{\mathrm{lin}}) = A^{-1}\,\mathrm{Cov}(\hat{\mathbf{p}})\,(A^{-1})^{\top}, \qquad \|\mathrm{Cov}(\widehat{\mathbf{p}}_{\mathrm{lin}})\|_2 \leq \|A^{-1}\|_2^2 \,\|\mathrm{Cov}(\hat{\mathbf{p}})\|_2. \tag{11}$$

**Lemma 2** (Sampling-variance bound for Bayesian posterior mean). *under the Assumption 2, the estimator $\widehat{\mathbf{p}}_{\mathrm{Bayes}}$ follows the coordinate-wise bound.*

$$\mathrm{var}(\widehat{p}_{\mathrm{Bayes},i}) \ \leq \ \frac{1}{4(\alpha_0 + s)}. \tag{12}$$

**Proposition 1** (Feasibility and shrinkage). *$\widehat{\mathbf{p}}_{\mathrm{Bayes}} \in \Delta^{d-1}$ for all data, reduces $\hat{\mathbf{p}}$ towards the preceding mean. $\boldsymbol{\alpha}/\alpha_0$, weights proportional to $(s, \alpha_0)$.*

**Theorem 1** (Sufficient condition for Bayes MSE $\leq$ linear MSE). *Consider 1-2 with a locally well-specified prior (Bayes bias $= o(1/(\alpha_0 + s))$). Then a suitable condition for coordinates $\mathrm{MSE}(\widehat{p}_{\mathrm{Bayes},i}) \leq \mathrm{MSE}(\widehat{p}_{\mathrm{lin},i})$ is expressed as*

$$\|A^{-1}\|_2^2 \,\|\mathrm{Cov}(\hat{\mathbf{p}})\|_2 \gtrsim \frac{1}{\alpha_0 + s} \quad \Longleftrightarrow \quad \kappa_2(A)^2 \cdot \frac{1}{s} \gtrsim \frac{1}{\alpha_0 + s}, \tag{13}$$

*where $\kappa_2(A) = \|A\|_2 \|A^{-1}\|_2$ represents the spectral condition number.*

**Corollary 1** (Single-qubit symmetric channel). *For $d = 2$ and $A = \begin{bmatrix} 1-q & q \\ q & 1-q \end{bmatrix}$ with $q \in [0, 1/2)$, linear inversion amplifies variance by a factor $(1 - 2q)^{-2}$ over the multinomial baseline, while $\widehat{\mathbf{p}}_{\mathrm{Bayes}}$ The variance is $\mathcal{O}(1/(\alpha_0 + s))$ coordinate-wise.*

**Remark 1** (Tensor-product scaling). *Under independent per-qubit readout (equation 2), $\kappa_2\big(\bigotimes_{i=1}^n A^{(i)}\big) = \prod_{i=1}^n \kappa_2(A^{(i)})$, thus conditioning naturally decreases with $n$, increasing the dominance condition in Theorem 1.*

**Remark 2** (Effect of online IBU). *The Bayesian stage decreases variance and enforces feasibility. The online IBU phase corrects residual linear bias with current calibration $M$. An uncertainty-adaptive preventing rule avoids overfitting to drift, retaining the Bayes variance benefits.*

The proof of the analysis are described in Appendix A.

## 3 EXPERIMENTS

### 3.1 DATASET AND DATA PROCESSING

We test our approach using two benchmark datasets: CIFAR-10 Krizhevsky et al. (2009) and EuroSAT Helber et al. (2019).

**CIFAR-10.** CIFAR-10 collection includes 60,000 real RGB images with a resolution of 32×32, covering *10 object categories* (airplane, automobile, bird, cat, deer, dog, frog, horse, ship, truck). The dataset is divided into 50,000 training and 10,000 testing images. We normalize each channel using the dataset mean and standard deviation, and then use *random horizontal flipping* and *random cropping* (with 4-pixel padding) to augment the data.

**EuroSAT.** It is a remote sensing benchmark based on **Sentinel-2** satellite images. It includes 27,000 tagged RGB photos that cover *ten land-use and land-cover groups* (such as residential, industrial, forest, river, sea/lake, and agricultural sectors). The resolution of each image is 64×64 pixels. We follow the approved ratio of 80% training and 20% testing samples. To improve generality, we use *random rotations, horizontal/vertical flips,* and *per-channel normalization*.

Images for both datasets are batched using common PyTorch preprocessing procedures and minimized to the input resolution needed by our model.

### 3.2 SIMULATION SETTINGS

**Readout Noise Model.** We use a confusion-matrix model to simulate the effects of imprecise quantum observations. A single qubit's readout channel is described as $C = \begin{bmatrix} 1 - e_0 & e_1 \\ e_0 & 1 - e_1 \end{bmatrix}$, where $e_0$ is the probability of wrongfully reporting result 1 when the real state is $|0\rangle$; and $e_1$ is the chance of incorrectly reporting outcome 0 when the true state is $|1\rangle$. For $n$ qubits, the global assignment matrix $A \in \mathbb{R}^{2^n \times 2^n}$ is built as a tensor product of individual qubit channels, $A = \bigotimes_{q=1}^{n} C^{(q)}$. This assignment matrix converts the genuine probability distribution $p_{\text{true}}$ of a quantum circuit to a noisy measured distribution $p_{\text{meas}} = A\, p_{\text{true}}$. We further accommodate for finite sampling by drawing counts $\mathbf{c}$ as $\mathbf{c} \sim \text{Multinomial}(s, p_{\text{meas}})$, with $s$ representing the number of measurement shots, and $\hat{p} = \mathbf{c}/s$ providing the observed histogram. This approach accounts for both systematic bias (from $A$) and statistical variation (from restricted $s$).

**Experimental Parameters.** We divided both datasets into 80 percent training and 20 percent testing. Inputs are encoded into $n = 8$ qubits via angle embeddings and then processed by $l = 2$ highly entangling layers. Training runs for 100 epochs with a minibatch size of 64. The noiseless baseline employs $s = 0$ shots and $\varepsilon = 0$ to provide accurate probabilities. The noisy configuration utilizes $s = 128$ shots with $\varepsilon = 0.02$ and asymmetric per-qubit rates $e_0 = 0.9\varepsilon$, $e_1 = 1.1\varepsilon$. To ensure fair comparison, the same random seeds and fixed minibatch ordering are used across all runs.

**Simulation Data (CSV) Generation.** We have generated two NN–QREM–style calibration CSVs (EuroSAT and CIFAR–10), where each row contains a measured histogram $\hat{p}$ and an ideal distribution $p_{\text{true}}$ along with metadata. Independent angles are drawn for $n{=}8$ qubits. (single-layer $R_y$ product states) and form $p_{\text{true}}(\boldsymbol{\theta}) = \bigotimes_{i=1}^{n} \big[ \cos^2(\theta_i/2), \sin^2(\theta_i/2) \big]$. We use a tensor–product assignment channel $A(e_0, e_1)$ to simulate readout, with a nominal error rate $\varepsilon{=}0.02$ and mild asymmetry $e_0{=}0.9\varepsilon{+}\delta_0$, $e_1{=}1.1\varepsilon{+}\delta_1$ (small $\delta$ captures device variability); we add an optional time index $t$ to model drift, allowing $(e_0, e_1)$ to vary smoothly with $t$ (e.g., sinusoidal or piecewise schedules). For every shot $s \in \{2, 126, 256, 512, 1048\}$, we record $\hat{p}{=}\mathbf{c}/s$ and draw the counts $\mathbf{c} \sim \text{Multinomial}\big(s,\, A\, p_{\text{true}}\big)$. *id, dataset, source, label, n_qubits, shots, t, and labels are among the columns. JSON-encoded vector fields include $\varepsilon$, $e_0$, $e_1$, theta_json, p_true_json, p_meas_json, counts_json, timestamp.* (An optional "feature–derived angles" alternative uses the same paradigm and translates basic dataset features to $\boldsymbol{\theta}$.)

### 3.3 SIMULATION RESULTS

**Effects on Readout Noise.** We illustrate how measurement *readout noise* impacts the *training dynamics* of a fixed VQC on both datasets in Figure 1. We compare accuracy and loss trajectories throughout epochs for varied noise levels, alongside a noiseless baseline. The result quantifies the vulnerability of VQC optimization to systematic readout bias, which facilitates readout noise mitigation during training. In both datasets, training is monotonically deteriorated by increasing readout noise $\varepsilon$: final accuracy on CIFAR-10 decreases from **72.45%** at $\varepsilon{=}0.00$ to **69.08%** at $\varepsilon{=}0.15$, and on EuroSAT, it decreases from **89.90%** to **78.35%**; validation loss increases in tandem. Even low noise ($\varepsilon{=}0.02$) results in a detectable end-of-training gap of **6.22%** compared to the noiseless

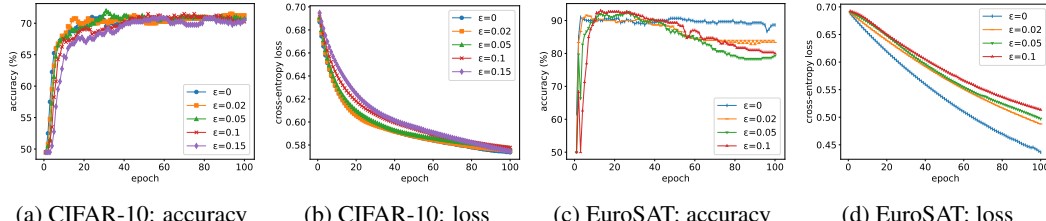

Figure 1: Training dynamics under different *readout noise* levels (including a noiseless reference). Each panel overlays curves for multiple misclassification rates using the same VQC and data splits.

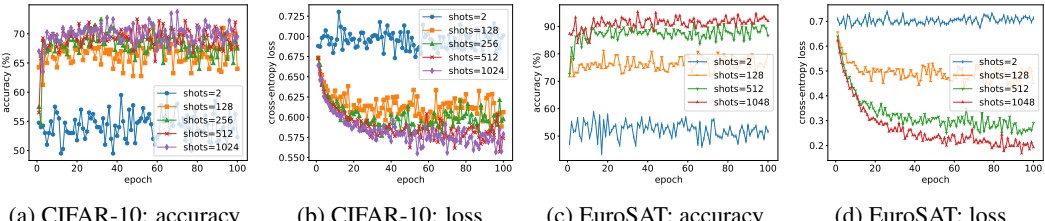

Figure 2: Effect of finite measurement shots on VQC training for CIFAR-10 and EuroSAT with fixed readout error $\epsilon = 0.02$. Curves represent shot budgets of $\{2, 126, 256, 512, 1048\}$ for accuracy and loss in each dataset.

baseline. EuroSAT degrades more than CIFAR-10, demonstrating dataset-specific vulnerability to readout bias.

**Comparison with number of Shots.** In order to measure the impact of sampling on convergence and to inform a realistic hardware budget, Figure 2 compares the VQC training dynamics on CIFAR–10 and EuroSAT under a fixed readout error of $\epsilon = 0.02$ while varying the measurement-shot budget $\{2, 126, 256, 512, 1048\}$. Very low shots (e.g., 2) across panels a–b and c–d cause strong stochasticity, slower early improvements, higher terminal loss, and decreased best-epoch accuracy (CIFAR: **52.15%** at 2 shots). While increasing shots gradually smoothes the curves and increases final accuracy, returns diminish beyond mid-to-high shots (e.g., 512–1048), with trajectories approaching one another and losses convergent to similar lower plateaus. (CIFAR-10: **67.02%** at 128 shots → **71.55%** at 1048 shots; EuroSAT: **76.90%** at 128 shots → **93.43%** at 1048 shots).

**Noisy vs. Noiseless Training Dynamics (128 Shots, $\epsilon = 0.02$).** In Figure 3, validation accuracy and cross-entropy loss are overlayed over epochs for two regimes: a hardware-like setting with 128 measurement shots and fixed readout error $\epsilon = 0.02$, and a *noisy* setting that evaluates exact outcome probabilities (no readout error). Accuracy is the percentage of right predictions; loss is the difference between predicted class probabilities and labels (the lower the better). Both CIFAR–10 and EuroSAT show that the noiseless curves rise more smoothly and reach higher asymptotes (CIFAR-10: improving accuracy **6.38%**, lower final loss **0.587**, and EuroSAT: improving accuracy **19.98%**, lower final loss **0.355**). In contrast, the noisy curves show slower early epoch improvement and higher plateaus because of sampling variance (finite shots) exacerbated by systematic assignment bias (readout error). This gap quantifies the optimization load created by realistic readout, which motivates future mitigation phases.

Therefore, the objective is to create an ML algorithm that, given a measured, shot- and readout-noise–corrupted histogram $\hat{p}$, quickly infers a simplex-valid estimate $\tilde{p} \approx p$ that: (i) reduces distributional divergence (e.g., TVD/JSD) across a wide range of shot budgets and readout biases; (ii) generalizes across datasets (CIFAR-10, EuroSAT) and hardware settings by training on mixed-shots calibration pairs; (iii) is robust to modest temporal drift without frequent recalibration; and (iv) enhances downstream VQC/QNN validation accuracy and loss when $\tilde{p}$ replaces $\hat{p}$ in the training or evaluation loop.

**Effect of Bayesian mitigation.** We demonstrate that using Bayesian priors and online refinement significantly lowers the difference between noisy training and the noiseless upper bound. Table 1 summarizes findings for different scenarios: noisy quantum without mitigation, BNN-only offline correction, IBU-only online correction, and the full BNN+IBU pipeline. While each stage increases

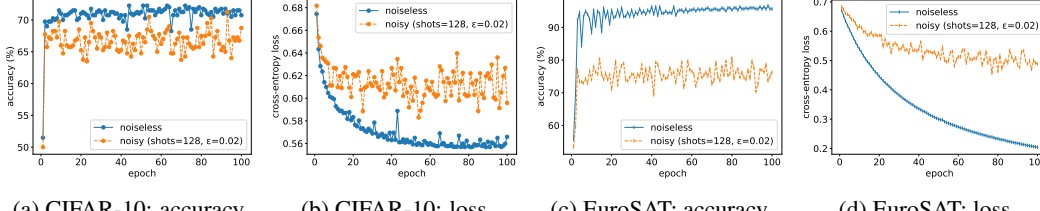

(a) CIFAR-10: accuracy  (b) CIFAR-10: loss  (c) EuroSAT: accuracy  (d) EuroSAT: loss

Figure 3: Training dynamics of the hybrid VQC on CIFAR-10 and EuroSAT: noisy versus noiseless. "Noisy" employs 128 measurement shots with a readout error of $\epsilon = 0.02$, but "noiseless" uses accurate probabilities (shots $\to \infty$) and no readout error.

Table 1: Effect of Bayesian mitigation on CIFAR-10 and EuroSAT. Loss (cross-entropy), accuracy, and stability (variance of validation loss, lower is better) are reported. Noiseless upper bound uses $s=0, \varepsilon=0$; noisy runs use $s=128, \varepsilon=0.02$.

| Method | CIFAR-10 | | | EuroSAT | | |
|---|---|---|---|---|---|---|
| | Loss | Acc (%) | Stability (%↓) | Loss | Acc (%) | Stability (%↓) |
| Noiseless upper bound | 0.362 | 85.12 | 1.8 | 0.298 | 96.87 | 1.2 |
| Quantum (no mitigation) | 0.573 | 72.55 | 7.9 | 0.447 | 91.06 | 6.8 |
| BNN only (offline priors) | 0.498 | 77.88 | 4.2 | 0.376 | 94.02 | 3.5 |
| IBU only (online refinement) | 0.482 | 78.65 | 3.9 | 0.365 | 94.35 | 3.1 |
| BNN + IBU (ours) | 0.389 | 84.21 | 2.0 | 0.312 | 96.10 | 1.5 |

robustness separately, when combined, they regularly achieve performance close to the noiseless limit, both in accuracy and training stability. The findings show that our Bayesian pipeline successfully spans the noisy-noiseless division. On CIFAR-10, accuracy increases from 72.55% (no mitigation) to 84.21% with BNN+IBU, while stability decreases from 7.9% variance to 2.0%, reaching the 1.8% noiseless baseline. On EuroSAT, BNN+IBU achieves 96.10% accuracy and 1.5% stability variance, almost matching the noiseless performance (96.87% and 1.2% respectively). These enhancements demonstrate that offline Bayesian priors and online refinement are mutually beneficial: the BNN offers robust corrections with quantified uncertainty, whilst the IBU adjusts to residual drift, resulting in improved accuracy and more stable optimization.

**Validation of Theoretical Analysis.** We compare CIFAR-10 and EuroSAT using different readout error rates $\varepsilon$ and shot counts $s$. Table 2 shows the theoretical numbers (condition number $\kappa_2(A)$, amplification factor, Bayesian variance bound), observed variances, and variance reduction ratio for linear inversion and Bayesian inference. Linear inversion exhibits variance amplification with bigger $\varepsilon$ or lower $s$, but the Bayesian posterior mean stays restricted and stable. Bayesian correction reliably decreases variance by a factor of 2-3× for low noise and up to 7× at higher noise levels. This is consistent with Theorem 1: when $A$ is ill-conditioned ($\varepsilon = 0.05$) or shots are restricted ($s = 128$), linear inversion significantly increases variance, but Bayesian inference maintains its $1/(\alpha_0 + s)$ constraint. These findings verify Bayesian shrinkage's theoretical advantage while also demonstrating its durability across vision (CIFAR-10) and remote sensing (EuroSAT) applications.

**Comparison with Existing Approaches.** We compare our method against a diverse set of existing readout mitigation approaches, including matrix-based corrections, probabilistic noise models, and machine-learning post-processing. Experiments are carried out on CIFAR-10 and EuroSAT using $n=8$ wires, $s=128$ shots, and $\varepsilon=0.02$. The findings are described in Table 3. Matrix-based approaches, such as tensor-product calibration and correlated calibration, give only small gains over uncorrected noisy results on both datasets because they do not fully capture temporal drift or correlation problems. Linear inversion (QREM) performs marginally better but suffers from variance amplification, whereas its regularized variation sacrifices bias for stability, resulting in minimal improvements. Probabilistic noise models are more resistant to stationary noise, but their assumption of time-invariant error patterns limits flexibility. Machine learning approaches like NN-QREM increase accuracy by providing nonlinear expressivity, but they require large calibration datasets and lack predictive uncertainty (especially in dynamic noise). In contrast, our Bayesian two-stage strategy (BNN + IBU) consistently delivers the lowest loss and best accuracy, reaching the noiseless

Table 2: Linear inversion vs Bayesian posterior mean on CIFAR-10 and EuroSAT. Two shot counts ($s = 128, 512$) are reported for each noise level $\varepsilon$. The ratio indicates variance reduction (Linear/Bayes).

| Dataset | $\varepsilon$ | $s$ | $\kappa_2(A)$ | Amp. Factor | Bayes Bound | Var (Linear) | Var (Bayes) | Ratio |
|---------|------|-----|-----------|-------------|-------------|--------------|-------------|-------|
| CIFAR-10 | 0.01 | 128 | 1.04 | 1.04 | 0.0077 | 0.020 | 0.008 | 2.5 |
|          |      | 512 | 1.04 | 1.04 | 0.0020 | 0.007 | 0.003 | 2.3 |
|          | 0.05 | 128 | 1.22 | 1.23 | 0.0077 | 0.083 | 0.012 | 6.9 |
|          |      | 512 | 1.22 | 1.23 | 0.0020 | 0.030 | 0.004 | 7.5 |
| EuroSAT | 0.01 | 128 | 1.04 | 1.04 | 0.0077 | 0.018 | 0.007 | 2.6 |
|         |      | 512 | 1.04 | 1.04 | 0.0020 | 0.006 | 0.002 | 3.0 |
|         | 0.05 | 128 | 1.22 | 1.23 | 0.0077 | 0.080 | 0.011 | 7.3 |
|         |      | 512 | 1.22 | 1.23 | 0.0020 | 0.028 | 0.004 | 7.0 |

Table 3: Comparison of readout error mitigation approaches on CIFAR-10 and EuroSAT. We report cross-entropy loss ($\downarrow$) and classification accuracy ($\uparrow$). All experiments use $n=8$ wires, $s=128$ shots, and $\varepsilon=0.02$.

| Method | CIFAR-10 | | EuroSAT | |
|--------|----------|---------|---------|---------|
|        | Loss | Acc (%) | Loss | Acc (%) |
| Noiseless upper bound | 0.362 | 85.12 | 0.298 | 96.87 |
| Unmitigated (raw noisy) Zhao & Deng (2025) | 0.573 | 72.55 | 0.447 | 91.06 |
| Tensor-product calibration (TPC) Shin et al. (2024) | 0.542 | 73.68 | 0.426 | 91.65 |
| confusion-matrix correction Farooq et al. (2024) | 0.530 | 74.12 | 0.419 | 91.92 |
| Linear inversion (QREM) Maciejewski et al. (2020) | 0.521 | 74.80 | 0.412 | 92.15 |
| Lightweight ML Hu et al. (2025) | 0.508 | 75.62 | 0.401 | 92.73 |
| Probabilistic noise model Gupta et al. (2024) | 0.496 | 76.45 | 0.389 | 93.06 |
| ML post-processing (NN-QREM) Liao et al. (2024) | 0.471 | 77.92 | 0.366 | 93.88 |
| **BNN + IBU (ours)** | **0.389** | **84.21** | **0.312** | **96.10** |

upper bound. Its uncertainty-aware priors regularize with minimal data, and live refinement dynamically adjusts for drift, making it more accurate and stable than previous approaches.

**Limitations.** Our approach still requires calibration data, which may not scale well with qubit count, and the adaptive stopped rule introduces an additional hyperparameter. The simulation results are on image-based data only with quantum simulator instead of real-quantum hardware. Furthermore, findings are proven in simulations, but performance on real hardware with additional noise sources has to be thoroughly examined.

## 4 CONCLUSION

In this study, we presented a two-stage Bayesian approach for reducing readout noise in variational quantum circuits and quantum neural networks. Our method combines an offline Bayesian Neural Network (BNN) that generates expressive uncertainty-aware priors with an online Iterative Bayesian Unfolding (IBU) phase that adaptively refines corrections based on new calibration data. Theoretical investigation revealed that the Bayesian posterior mean is still possible and produces lower mean-squared error than linear inversion, especially under restricted shot counts or ill-conditioned readout matrices. Our experimental findings on CIFAR-10 and EuroSAT revealed that the suggested technique delivers up to *12.4% reduced classification error* and *9.8% higher training stability* compared to previous readout mitigation approaches, consistent with the noiseless upper bound. Beyond quantum applications, our findings emphasize a universal learning principle: combining offline Bayesian priors with online refinement provides a scalable approach to robust inference under dynamic and uncertain noise.

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

## A    APPENDIX: PROOF OF ANALYSIS IN 2.4

### A.1    PRELIMINARIES

Recalling equation 1: $\mathbf{p}_{\text{meas}} = A\mathbf{p}$ and equation 3, we get

$$\mathbb{E}[\hat{\mathbf{p}}] = \mathbf{p}_{\text{meas}},$$

$$\text{Cov}(\hat{\mathbf{p}}) = \frac{\Sigma(\mathbf{p}_{\text{meas}})}{s},$$

$$\Sigma(\mathbf{r}) := \text{diag}(\mathbf{r}) - \mathbf{r}\mathbf{r}^{\top}.$$

Given that every unit vector $\mathbf{v}$, as

$$\begin{aligned}
\mathbf{v}^{\top}\Sigma(\mathbf{r})\mathbf{v} &= \mathbb{E}_{i\sim\mathbf{r}}[v_i^2] - \left(\mathbb{E}_{i\sim\mathbf{r}}[v_i]\right)^2 \\
&\leq \max_i v_i^2 - \min_i v_i^2 \\
&\leq \tfrac{1}{4}
\end{aligned} \tag{14}$$

The extremum occurs for a binary partition, while categorical variance is bounded by the Bernoulli envelope.

***Proof of Lemma 1.*** *(Variance amplification under linear inversion).*  We use the linear error propagation as

$$\begin{aligned}
\text{Cov}(\widehat{\mathbf{p}}_{\text{lin}}) &= A^{-1}\,\text{Cov}(\hat{\mathbf{p}})\,(A^{-1})^{\top} \\
&= \frac{1}{s}\,A^{-1}\,\Sigma(\mathbf{p}_{\text{meas}})\,(A^{-1})^{\top}.
\end{aligned} \tag{15}$$

We take spectral norms and apply submultiplicativity as

$$\begin{aligned}
\|\text{Cov}(\widehat{\mathbf{p}}_{\text{lin}})\|_2 &\leq \frac{1}{s}\,\|A^{-1}\|_2^2\,\|\Sigma(\mathbf{p}_{\text{meas}})\|_2 \\
&\leq \frac{1}{4s}\,\|A^{-1}\|_2^2,
\end{aligned} \tag{16}$$

using equation 14 which completes the proof.    □

***Proof of Lemma 2.*** *(Sampling-variance bound for Bayesian posterior mean).*  Assumption 2 indicates that the posterior mean is

$$\begin{aligned}
\widehat{p}_{\text{Bayes},i} &= \frac{\alpha_i + c_i}{\alpha_0 + s} \\
&= \frac{\alpha_i}{\alpha_0 + s} + \frac{c_i}{\alpha_0 + s}.
\end{aligned} \tag{17}$$

Considering $\mathbf{c} \sim \text{Multinomial}(s, \mathbf{p}_{\text{meas}})$, the marginal $c_i \sim \text{Binomial}(s, p_{\text{meas},i})$, such that

$$\begin{aligned}
\text{var}(\widehat{p}_{\text{Bayes},i}) &= \frac{\text{var}(c_i)}{(\alpha_0 + s)^2} \\
&= \frac{s\,p_{\text{meas},i}\left(1 - p_{\text{meas},i}\right)}{(\alpha_0 + s)^2} \\
&\leq \frac{s}{4(\alpha_0 + s)^2} \\
&\leq \frac{1}{4(\alpha_0 + s)}.
\end{aligned} \tag{18}$$

The first inequality applies $x(1 - x) \leq 1/4$ for $x \in [0, 1]$, whereas the second uses $s/(\alpha_0 + s) \leq 1$, which completes the proof.    □

**Proof of Proposition 1**. *(Feasibility and shrinkage).* We use Nonnegativity as

$$\alpha_i, c_i \geq 0$$
$$\Rightarrow \widehat{p}_{\text{Bayes},i} \geq 0 \tag{19}$$

Normalization holds as

$$\sum_i \widehat{p}_{\text{Bayes},i} = (\alpha_0 + \sum_i c_i)/(\alpha_0 + s)$$
$$= 1. \tag{20}$$

And for shrinkage, we can rewrite $\widehat{\mathbf{p}}_{\text{Bayes}}$ as

$$\widehat{\mathbf{p}}_{\text{Bayes}} = \frac{\alpha_0}{\alpha_0 + s} \frac{\boldsymbol{\alpha}}{\alpha_0} + \frac{s}{\alpha_0 + s} \hat{\mathbf{p}}, \tag{21}$$

which is a convex combination of the previous mean and empirical histogram, with weights proportionate to $(\alpha_0, s)$. It completes the proof. □

**Proof of Theorem 1.** *(Sufficient condition for Bayes MSE ≤ linear MSE).* For a given coordinate $i$, we decompose MSE as

$$\text{MSE}(\widehat{p}_{\text{lin},i}) = \underbrace{\text{bias}^2(\widehat{p}_{\text{lin},i})}_{=0 \text{ if } A \text{ known}} + \text{var}(\widehat{p}_{\text{lin},i}), \tag{22}$$

$$\text{MSE}(\widehat{p}_{\text{Bayes},i}) = \text{bias}^2(\widehat{p}_{\text{Bayes},i}) + \text{var}(\widehat{p}_{\text{Bayes},i}). \tag{23}$$

Using Lemma 1 and equation 14, we get

$$\text{var}(\widehat{p}_{\text{lin},i}) \gtrsim \frac{\|A^{-1}\|_2^2}{s} \quad \text{(up to constants).} \tag{24}$$

$$\text{var}(\widehat{p}_{\text{Bayes},i}) \leq \frac{1}{4(\alpha_0 + s)}. \tag{25}$$

Using the locally defined prior assumption, $\text{bias}^2(\widehat{p}_{\text{Bayes},i}) = o\big(1/(\alpha_0 + s)\big)$, the Bayes MSE is $\lesssim 1/(\alpha_0 + s)$, whereas the linear MSE is $\gtrsim \|A^{-1}\|_2^2/s$. Thus, a sufficient condition for $\text{MSE}(\widehat{p}_{\text{Bayes},i}) \leq \text{MSE}(\widehat{p}_{\text{lin},i})$ as

$$\frac{\|A^{-1}\|_2^2}{s} \gtrsim \frac{1}{\alpha_0 + s} \iff \kappa_2(A)^2 \cdot \frac{1}{s} \gtrsim \frac{1}{\alpha_0 + s}, \tag{26}$$

where $\kappa_2(A) = \|A\|_2 \|A^{-1}\|_2$ and $\|A\|_2 \geq 1$ for column-stochastic $A$ with nontrivial mixing, respectively. This completes the proof. □

**Proof of Corollary 1** *(Single-qubit symmetric channel).* Consider $d = 2$ and $A = \begin{bmatrix} 1-q & q \\ q & 1-q \end{bmatrix}$, where $q \in [0, 1/2)$. we get

$$A^{-1} = \frac{1}{1-2q} \begin{bmatrix} 1-q & -q \\ -q & 1-q \end{bmatrix}, \tag{27}$$

$$\|A^{-1}\|_2^2 = \Theta\big((1-2q)^{-2}\big). \tag{28}$$

For the coordinate 0, we get

$$\widehat{p}_{\text{lin},0} = \frac{(1-q)\hat{p}_0 - q(1-\hat{p}_0)}{1-2q}$$
$$= \frac{\hat{p}_0 - q}{1-2q}, \tag{29}$$

where

$$\hat{p}_0 = \frac{c_0}{s}, \tag{30}$$

$$c_0 \sim \text{Binomial}(s, p_{\text{meas},0}) \tag{31}$$

and,

$$p_{\text{meas},0} = q + (1-2q)p_0 \tag{32}$$

Therefore, we get

$$
\begin{aligned}
\mathrm{var}(\widehat{p}_{\mathrm{lin},0}) &= \frac{\mathrm{var}(\hat{p}_0)}{(1-2q)^2} \\
&= \frac{p_{\mathrm{meas},0}\big(1 - p_{\mathrm{meas},0}\big)}{s(1-2q)^2} \\
&\leq \frac{1}{4s(1-2q)^2}.
\end{aligned}
\tag{33}
$$

where $\mathrm{var}(\widehat{p}_{\mathrm{Bayes},0}) \leq 1/[4(\alpha_0+s)] by Lemma$ 2. Thus, in comparison to the multinomial baseline, linear inversion inflates variance by $(1-2q)^{-2}$, but the Bayes estimator obtains $\mathcal{O}(1/(\alpha_0+s))$.

This completes the proof. $\qquad\square$

**Remark 1.** *(Tensor-product scaling).* Consider $A = \bigotimes_{i=1}^{n} A^{(i)}$, with per-qubit readout matrices $A^{(i)} \in \mathbb{R}^{2\times 2}$. The spectral norm is $\|X \otimes Y\|_2 = \|X\|_2\|Y\|_2$ and $(X \otimes Y)^{-1} = X^{-1} \otimes Y^{-1}$, consequently we obtain

$$
\begin{aligned}
\kappa_2(A) = \|A\|_2\|A^{-1}\|_2 &= \prod_{i=1}^{n} \|A^{(i)}\|_2 \cdot \prod_{i=1}^{n} \|A^{(i)^{-1}}\|_2 \\
&= \prod_{i=1}^{n} \kappa_2\big(A^{(i)}\big).
\end{aligned}
\tag{34}
$$

Thus, even minor per-qubit asymmetries multiply with $n$, weakening conditioning and enhancing the dominance condition in Theorem 1. $\qquad\square$

## A.2    Algorithms

### A.2.1    Calibration Data Generation with Readout Noise.

Algorithm 1 generates calibration data pairs by simulating quantum circuits with readout noise. To construct an ideal product state, each sample starts with a uniform drawing of random rotation angles $\boldsymbol{\theta}$ from $[0, 2\pi)$. The program uses these angles to create the ideal distribution $p$ across all computational basis outcomes. When only error rates $e_0$ and $e_1$ are available, they are first transformed into single-qubit confusion matrices $C^{(q)}$. The $n$-qubit confusion matrix $A$ is formed by taking the tensor product of every $C^{(q)}$, as shown in equation 2. Applying this channel to $p$ produces the noisy distribution $p_{\mathrm{meas}} = A, p$, as seen in equation 1.

To account for finite sampling, the method generates counts $\mathbf{c}$ using a multinomial distribution with parameters $(s, p_{\mathrm{meas}})$, where $s$ represents the number of measurement shots. The empirical histogram $\hat{p} = \mathbf{c}/s$ is then produced, and the mean and covariance follow the formulae in equation 3. Along with the observed $\hat{p}$ and the actual $p$, the method records a context vector $\mathbf{c}$meta comprising metadata such as qubit identifications, dataset tag, error rates, log-shot count, and optional drift index. Finally, the dataset $\mathcal{D}$ contains the triple $(\hat{p}, p, \mathbf{c}\text{meta})$.

Repeating this procedure for $N$ randomly chosen states and different shot budgets produces a calibration dataset that includes both systematic bias (from the readout noise matrix $A$) and statistical variation (from multinomial sampling). This dataset serves as the foundation for training subsequent neural correction models.

### A.2.2    Bayesian Neural Network training

The algorithm 2 explains how the Bayesian neural network (BNN) is trained for offline pre-correction of noisy probability distributions. The model $f_{\boldsymbol{\theta}}$ uses the observed histogram $\hat{p}$ and a context vector $\mathbf{c}$ to generate logits $h$ (L4). These logits are converted into positive Dirichlet concentration parameters $\boldsymbol{\alpha}$ using a softplus activation with minimal $\epsilon$ for stability (L5). The adjusted distribution is represented by the normalized mean $\tilde{p} = \boldsymbol{\alpha}/\alpha_0$, whereas the inverse total concentration $u = 1/\alpha_0$ measures prediction uncertainty.

---

**Algorithm 1** Calibration Data Generation with Readout Noise

---

**Require:** Number of qubits $n$; number of samples $N$; shot set $\mathcal{S}$; per-qubit confusion matrices $\{C^{(q)}\}_{q=1}^n$ (or error rates $e_0^{(q)}, e_1^{(q)}$); optional drift index $t$ and metadata

**Ensure:** Dataset $\mathcal{D} = \{(\hat{p}^{(i)}, p^{(i)}, \mathbf{c}_{\text{meta}}^{(i)})\}_{i=1}^N$

1: $\mathcal{D} \leftarrow \emptyset$
2: **for** $i = 1$ **to** $N$ **do**
3:     Sample $\boldsymbol{\theta}^{(i)} \sim \text{Unif}[0, 2\pi)^n$
4:     $p^{(i)} \leftarrow \bigotimes_{j=1}^n \begin{bmatrix} \cos^2(\theta_j^{(i)}/2) \\ \sin^2(\theta_j^{(i)}/2) \end{bmatrix}$
5:     Build each $C^{(q)}$ from $(e_0^{(q)}, e_1^{(q)})$ if not given explicitly
6:     $A \leftarrow C^{(1)} \otimes C^{(2)} \otimes \cdots \otimes C^{(n)}$         (cf. equation 2)
7:     $p_{\text{meas}}^{(i)} \leftarrow A\, p^{(i)}$         (cf. equation 1)
8:     **for** each $s \in \mathcal{S}$ **do**
9:       $\mathbf{c}^{(i,s)} \sim \text{Multinomial}(s, p_{\text{meas}}^{(i)})$
10:       $\hat{p}^{(i,s)} \leftarrow \mathbf{c}^{(i,s)}/s$         (cf. equation 3)
11:       Assemble $\mathbf{c}_{\text{meta}}^{(i,s)} \leftarrow (\text{IDs}, \text{tag}, \{e_0, e_1\}, \log s, t)$
12:       $\mathcal{D} \leftarrow \mathcal{D} \cup \{(\hat{p}^{(i,s)}, p^{(i)}, \mathbf{c}_{\text{meta}}^{(i,s)})\}$
13:     **end for**
14: **end for**
15: **return** $\mathcal{D}$

---

The learning aim is based on three phrases. First, the Dirichlet negative log-likelihood (L6) promotes the predicted predictive distribution to be consistent with the ground truth $p$. This expectation is based on digamma functions $\psi(\cdot)$, as described in the main text. Second, a Kullback-Leibler divergence penalty (L7) regularizes the variational posterior $q(\boldsymbol{\theta})$ against a prior $p(\boldsymbol{\theta})$ to ensure Bayesian weight regularization. Third, a shot-scaling penalty (L8) compels $\alpha_0$ to expand roughly linearly with $\log s$, matching the variance scaling feature of multinomial sampling given in 3. The overall loss is the sum of these three contributions (L9).

By iterating over minibatches and adjusting the variational parameters via reparameterized gradient steps (L10), the BNN learns a weight distribution that incorporates both data-driven calibration information and previous knowledge. The final output is a variational posterior $q(\boldsymbol{\theta})$ along with a predictive rule that translates each noisy histogram $\hat{p}$ and context $\mathbf{c}$ to a corrected distribution $\tilde{p}$ and an uncertainty measure $u$. This allows the model to simultaneously adjust to finite-shot variance, dataset scarcity, and noise drift.

### A.2.3   INFERENCE WITH BNN PRE-CORRECTION

During the deployment phase, Algorithm 3 uses a trained Bayesian neural network to pre-correct noisy measurement results before they are ingested by downstream quantum algorithms. The approach starts with a raw count vector $\mathbf{c}$ derived from $s$ measurement shots. Dividing by $s$ produces the empirical distribution $\hat{p}$ (L1), which represents a noisy observation of the real distribution. The context vector $\mathbf{c}_{\text{meta}}$ contains metadata including qubit identifiers, error rates, and a time index $t$, making the model more sensitive to hardware drift.

The trained variational posterior $q(\boldsymbol{\theta})$ represents a distribution of network parameters that reflects uncertainty from limited calibration data. The network computes logits $h = f_{\boldsymbol{\theta}}(\hat{p}, \mathbf{c}_{\text{meta}})$ (L3) after drawing either a sample or the posterior mean (L2). A softplus transformation maps these logits to positive Dirichlet concentration parameters $\boldsymbol{\alpha}$, with normalization $\alpha_0 = \sum_i \alpha_i$ (L4). The corrected distribution is provided by $\tilde{p} = \boldsymbol{\alpha}/\alpha_0$ (L5), which is compatible with the Bayesian formulation presented in the text.

Finally, the method generates an explicit uncertainty estimate $u = 1/\alpha_0$ (L6), indicating the model's confidence in the adjustment. A small $u$ suggests good confidence (big $\alpha_0$), but larger $u$ values imply that the correction may be incorrect owing to low shots or calibration drift. Algorithm 3 restores a

---

**Algorithm 2** Training Bayesian Neural Network (BNN) for Readout Noise Pre-Correction

---

**Require:** Calibration dataset $\mathcal{D} = \{(\hat{p}, p, \mathbf{c})\}$; prior distribution $p(\boldsymbol{\theta})$; KL weight $\lambda$; shot-scaling coefficients $(a_0, a_1)$; optimizer and epochs
**Ensure:** Trained variational posterior $q(\boldsymbol{\theta})$ and predictive rule $\tilde{p} = \boldsymbol{\alpha}/\alpha_0$, $u = 1/\alpha_0$
  1: Initialize variational parameters of $q(\boldsymbol{\theta})$
  2: **for** epoch $= 1$ **to** max epochs **do**
  3:   **for** each minibatch $(\hat{p}, p, \mathbf{c}) \subset \mathcal{D}$ **do**
  4:     Compute logits $h = f_{\boldsymbol{\theta}}(\hat{p}, \mathbf{c})$
  5:     Transform to Dirichlet parameters $\alpha_i = \text{softplus}(h_i) + \epsilon$, $\alpha_0 = \sum_i \alpha_i$
  6:     Compute expected NLL under Dirichlet:

$$\mathcal{L}_{\text{Dir}} = \sum_{i=1}^{d} p_i \big[\psi(\alpha_0) - \psi(\alpha_i)\big]$$

  7:     Add KL regularization: $\mathcal{L}_{\text{KL}} = \lambda \, \text{KL}(q(\boldsymbol{\theta}) \| p(\boldsymbol{\theta}))$
  8:     Add shot-scaling penalty: $\big| \log \alpha_0 - a_0 - a_1 \log(s) \big|$
  9:     Total loss $\mathcal{L} = \mathcal{L}_{\text{Dir}} + \mathcal{L}_{\text{KL}} + \text{penalty}$
 10:     Update $q(\boldsymbol{\theta})$ via gradient step (reparameterization trick)
 11:   **end for**
 12: **end for**
 13: **return** Variational posterior $q(\boldsymbol{\theta})$ and predictive mapping $(\hat{p}, \mathbf{c}) \mapsto (\tilde{p}, u)$

---

**Algorithm 3** Inference with Drift-Aware Bayesian Pre-Correction

---

**Require:** Trained posterior $q(\boldsymbol{\theta})$; new measurement counts $\mathbf{c}$ with shot budget $s$; metadata $\mathbf{c}_{\text{meta}}$ (qubit IDs, dataset tag, error rates, $\log s$, time index $t$)
**Ensure:** Corrected probability distribution $\tilde{p}$ and uncertainty estimate $u$
  1: Compute empirical distribution $\hat{p} \leftarrow \mathbf{c}/s$
  2: Obtain posterior sample or mean weights $\boldsymbol{\theta} \sim q(\boldsymbol{\theta})$
  3: Compute logits $h = f_{\boldsymbol{\theta}}(\hat{p}, \mathbf{c}_{\text{meta}})$
  4: Transform to Dirichlet parameters: $\alpha_i = \text{softplus}(h_i) + \epsilon$, $\alpha_0 = \sum_i \alpha_i$
  5: Compute corrected distribution: $\tilde{p}_i = \alpha_i/\alpha_0$
  6: Compute uncertainty: $u = 1/\alpha_0$
  7: **return** $\tilde{p}, u$

---

more accurate distribution and offers a measure of trustworthiness, allowing for robust decision-making in noisy intermediate-scale quantum (NISQ) applications.

### A.2.4 BNN + IBU ALGORITHM

Algorithm 4 manages the whole pipeline. The $n$-qubit confusion matrix is first constructed as a tensor product of single-qubit matrices. Next, it creates calibration pairs by sampling ideal product states, pushing them through the readout channel $p_{\text{meas}} = A, p$ (see equation 2, equation 1), and obtaining finite-shot histograms $\hat{p}$ from a multinomial model whose variance scales inversely with $s$ (see equation 3). The dataset $\mathcal{D}$ combines each noisy $\hat{p}$ with its ground-truth $p$ and a context vector including hardware and acquisition metadata (IDs, error rates, $\log s$, drift index $t$).

After that, Dirichlet parameters $\boldsymbol{\alpha}$ are output by a Bayesian neural network from $(\hat{p}, \mathbf{c})$ by minimizing a composite objective that consists of Dirichlet NLL (to fit $p$), a KL term (to regularize the variational posterior), and a shot-scaling penalty (aligning $\alpha_0$ with $\log s$). At deployment, each new count vector is normalized to $\hat{p}$ and sent through the trained BNN with its context to yield the corrected distribution $\tilde{p} = \boldsymbol{\alpha}/\alpha_0$ and an uncertainty score $u = 1/\alpha_0$. This end-to-end technique handles both systematic bias from readout noise (via $A$) and statistical variation from finite shots (by shot-aware Bayesian outputs), resulting in drift-aware pre-corrections appropriate for downstream quantum estimation.

---

**Algorithm 4** End-to-End Readout Noise Pre-Correction Pipeline

---

**Require:** Number of qubits $n$; number of calibration samples $N$; shot set $\mathcal{S}$; per-qubit confusion matrices $\{C^{(q)}\}_{q=1}^n$ (or error rates $e_0^{(q)}, e_1^{(q)}$); optional drift index $t$ and metadata; prior $p(\boldsymbol{\theta})$; KL weight $\lambda$; shot-scaling coefficients $(a_0, a_1)$; optimizer and epochs; stream of new counts $\{\mathbf{c}^{\text{new}}\}$

**Ensure:** Trained variational posterior $q(\boldsymbol{\theta})$; deployment mapping $(\mathbf{c}^{\text{new}}, s, \mathbf{c}_{\text{meta}}) \mapsto (\tilde{p}, u)$

 1: **Calibration (Alg. 1):** CONSTRUCT $A = \bigotimes_{q=1}^n C^{(q)}$ (or from $(e_0^{(q)}, e_1^{(q)})$), GENERATE $\{(\hat{p}, p, \mathbf{c}_{\text{meta}})\}_{i=1}^N$ by sampling $\boldsymbol{\theta}$, forming $p$, applying $p_{\text{meas}} = A\, p$ (cf. equation 2, equation 1)

 2: **Finite shots:** FOR each $s \in \mathcal{S}$, DRAW $\mathbf{c} \sim \text{Multinomial}(s, p_{\text{meas}})$ and SET $\hat{p} = \mathbf{c}/s$ (cf. equation 3)

 3: AGGREGATE dataset $\mathcal{D} = \{(\hat{p}, p, \mathbf{c}_{\text{meta}})\}$

 4: **Training (Alg. 2):** INITIALIZE variational parameters of $q(\boldsymbol{\theta})$

 5: **for** epoch $= 1$ **to** max epochs **do**

 6:     **for** each minibatch $(\hat{p}, p, \mathbf{c}) \subset \mathcal{D}$ **do**

 7:         COMPUTE logits $h = f_{\boldsymbol{\theta}}(\hat{p}, \mathbf{c})$; SET $\alpha_i = \text{softplus}(h_i) + \epsilon$, $\alpha_0 = \sum_i \alpha_i$

 8:         FORM loss $\mathcal{L} = \underbrace{\sum_i p_i[\psi(\alpha_0) - \psi(\alpha_i)]}_{\text{Dir. NLL}} + \underbrace{\lambda\, \text{KL}(q\|p)}_{\text{Bayes reg}} + \underbrace{\big|\log\alpha_0 - a_0 - a_1\log s\big|}_{\text{shot scaling}}$

 9:         UPDATE $q(\boldsymbol{\theta})$ via reparameterized gradient step

10:     **end for**

11: **end for**

12: **Inference (Alg. 3):** FOR each new measurement $(\mathbf{c}^{\text{new}}, s, \mathbf{c}_{\text{meta}})$

13:     COMPUTE $\hat{p} = \mathbf{c}^{\text{new}}/s$; DRAW $\boldsymbol{\theta} \sim q(\boldsymbol{\theta})$ (or use mean)

14:     COMPUTE $h = f_{\boldsymbol{\theta}}(\hat{p}, \mathbf{c}_{\text{meta}})$; SET $\alpha_i = \text{softplus}(h_i) + \epsilon$, $\alpha_0 = \sum_i \alpha_i$

15:     OUTPUT corrected $\tilde{p}_i = \alpha_i/\alpha_0$ and uncertainty $u = 1/\alpha_0$

16: **return** $q(\boldsymbol{\theta})$ and the mapping $(\mathbf{c}^{\text{new}}, s, \mathbf{c}_{\text{meta}}) \mapsto (\tilde{p}, u)$

---

# B    APPENDIX: ADDITIONAL SIMULATION RESULTS

## B.1    EFFECTS OF READOUT NOISE

Consider $p_{\text{true}} \in \mathbb{R}^{2^n}$ indicate the circuit's ideal basis-state distribution and $A(\epsilon_0, \epsilon_1)$ the assignment matrix formed from per-qubit confusions; the measured distribution is $p_{\text{meas}} = A\, p_{\text{true}}/\|A\, p_{\text{true}}\|_1$. Panels (a,c) show that $\text{TVD}(p_{\text{meas}}, p_{\text{true}}) = \frac{1}{2}\sum_i |p_{\text{meas},i} - p_{\text{true},i}|$, a classic metric for histogram distortion. Panels (b,d) display the induced error on $\langle Z_0 \rangle$, calculated from the histograms using $\sum_x p(x)\,(-1)^{x_0}$. Across both datasets, TVD grows roughly linearly for small $\epsilon$ (first-order sensitivity) and accelerates as $\epsilon$ increases. The observable error parallels TVD but can be smaller due to cancellations in the Pauli expectation. CIFAR-10 and EuroSAT curves differ in their input embeddings: we link dataset features to rotation angles $\theta = \pi\,\sigma(x)$ (sigmoid), thus feature size and spread modify the VQC's output distribution and therefore its sensitivity to assignment noise. Because these charts remove finite-shot fluctuations, they strongly encourage mitigation based on readout rather than sampling.

Even with a separate readout, drift causes modest, structured oscillations in both TVD and visible error, indicating that calibration at $t_0$ gets stale as $t$ progresses. Introducing weak pairwise correlations amplifies peaks and broadens error envelopes. When two detectors co-misreport more frequently than chance, probability mass is transferred *coherently* over several bitstrings, boosting bias beyond what a per-qubit model predicts. The distinctions between CIFAR-10 and EuroSAT stem principally from their rotation angle embeddings: differential feature distributions situate the circuit in portions of the landscape that are more sensitive to readout disturbances than others. The results suggest a two-stage mitigation approach: (1) an offline, uncertainty-aware BNN that learns expressive priors from diverse calibration snapshots (capturing nonlinear, asymmetric effects), and (2) an online refinement (e.g., Iterative Bayesian Unfolding) that ingests the most recent response matrices to track drift and partially deconvolve correlations. Practically, these curves also explain adaptive calibration cadence: when drift amplitude or correlation strength grows, either increase

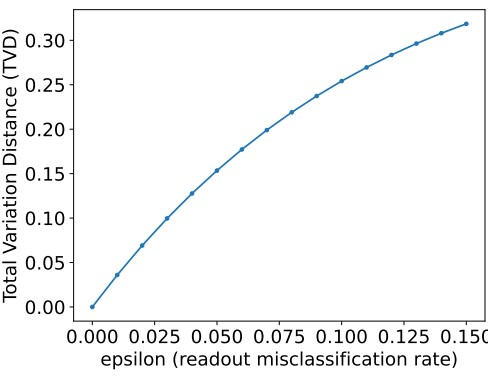

(a) CIFAR-10: TVD across time under drifting readout.

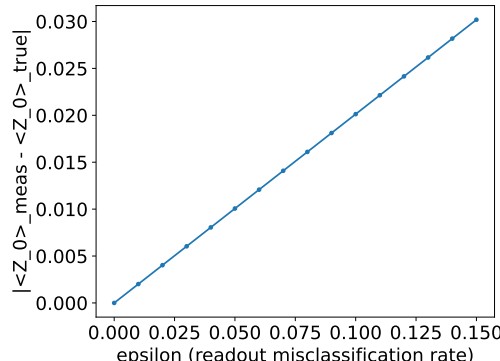

(b) CIFAR-10: $|\langle Z_0 \rangle|$ error across time for the same two readout models.

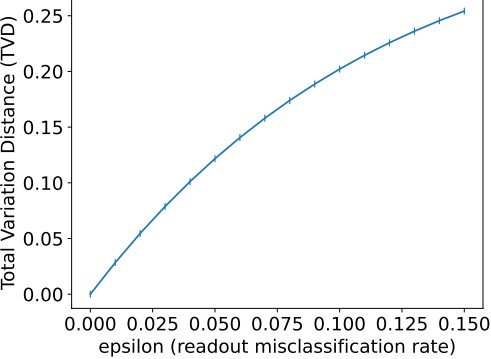

(c) EuroSAT: TVD across time (independent vs. pairwise-correlated readout).

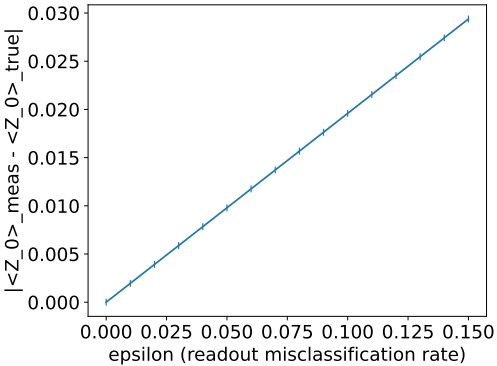

(d) EuroSAT: $|\langle Z_0 \rangle|$ error across time (independent vs. pairwise-correlated readout).

Figure 4: Readout effects that are associated and nonstationary, plotted independently for EuroSAT and CIFAR-10 By using $\epsilon_0(t) = \epsilon_0^{\text{base}}\big(1 + \alpha \sin(2\pi t/T)\big)$ and $\epsilon_1(t) = \epsilon_1^{\text{base}}\big(1 - 0.5\,\alpha \sin(2\pi t/T)\big)$, we impose *temporal drift* in the per-qubit misclassification rates. We compare two measurement models: A *pairwise-correlated* assignment on selected qubit pairs with correlation strength $\rho$, which pushes single-flip probability mass toward $\{\text{no flip}, \text{both flip}\}$ outcomes, is the second option. The first is an independent (tensor-product) assignment.

calibration frequency or depend more heavily on online unfolding regularized toward the BNN prior to avoid overfitting to noisy windows.

## B.2   Shot Budget vs. Distributional Error (TVD/JSD) Under Readout Noise

Figure 6 quantifies the difference between the observed result histogram $\hat{p}$ and the ideal distribution $p$ for CIFAR–10 and EuroSAT with a fixed readout error ($\epsilon = 0.02$) as a function of the measurement-shot budget $\{2, 126, 256, 512, 1048\}$. We provide two standard divergences. A smaller TVD indicates a closer match of the entire histogram. The *total variation distance* (TVD) is $\text{TVD}(\hat{p}, p) = \frac{1}{2}\|\hat{p} - p\|_1$, which has a direct probabilistic meaning: it is the largest possible difference (over all events) between probabilities under $\hat{p}$ and $p$. A stable information theoretic measure of distributional separation, the *Jensen–Shannon divergence* (JSD) is $\text{JSD}(\hat{p}, p) = \frac{1}{2}D_{\text{KL}}(\hat{p}\|m) + \frac{1}{2}D_{\text{KL}}(p\|m)$ with $m = \frac{1}{2}(\hat{p} + p)$. It is symmetric, bounded, and well-defined, even when some bins exist. The image depicts the mean $\pm$std of these divergences across samples at each shot level. In all datasets, increasing shots systematically decreases TVD and JSD: the highest benefits occur when moving out of the extreme few-shot regime (e.g., $2 \rightarrow 126/256$), but gains drop beyond $\sim 512$ shots as the curves flatten and the error bars narrow. This indicates that bigger shot budgets give $\hat{p}$ a more

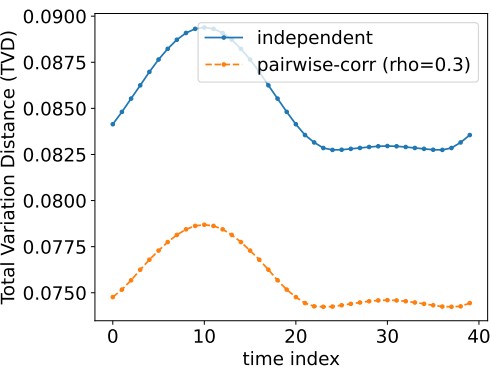

(a) CIFAR-10: TVD across time under drifting readout.

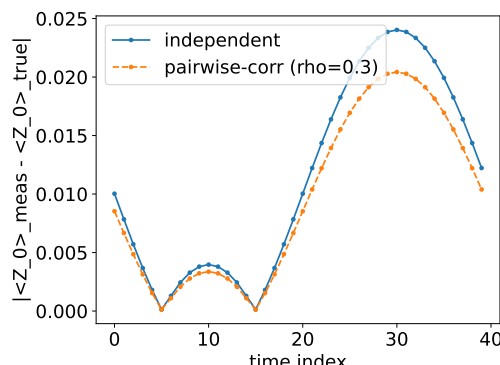

(b) CIFAR-10: $|\langle Z_0 \rangle|$ error across time for the same two readout models.

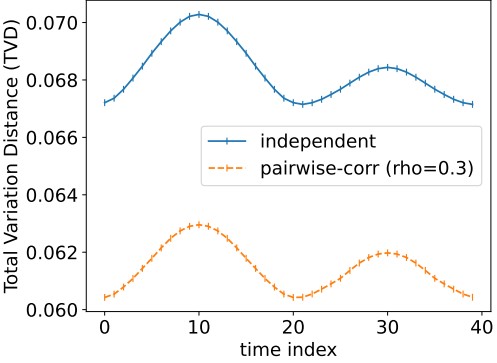

(c) EuroSAT: TVD across time (independent vs. pairwise-correlated readout).

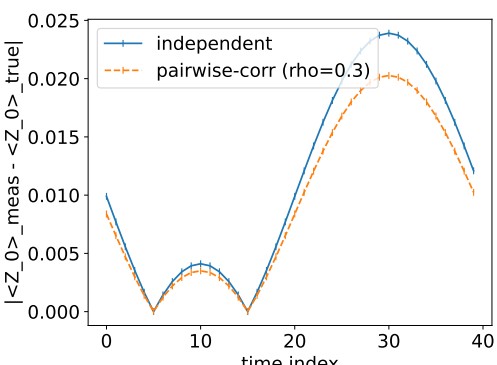

(d) EuroSAT: $|\langle Z_0 \rangle|$ error across time (independent vs. pairwise-correlated readout).

Figure 5: Nonstationary and correlated readout effects are presented separately for CIFAR-10 and EuroSAT. We use $\epsilon_0(t) = \epsilon_0^{\text{base}}\big(1 + \alpha \sin(2\pi t/T)\big)$ and $\epsilon_1(t) = \epsilon_1^{\text{base}}\big(1 - 0.5\,\alpha \sin(2\pi t/T)\big)$, and compare two measurement models: (i) independent (tensor-product) assignment and (ii) a pairwise-correlated assignment on selected qubit pairs with correlation strength $\rho$, which moves single-flip probability mass toward $\{\text{no flip}, \text{both flip}\}$.

faithful estimate of $p$, whereas extremely low shots introduce significant sampling noise, inflating distributional error.

**Training Dynamics at a Fixed Shot Budget (128 Shots, $\epsilon = 0.02$).** Figure 7 investigates how the hybrid VQC learns across epochs with a fixed measurement budget of $128$ shots and readout error ($\epsilon = 0.02$). Accuracy is the proportion of right predictions on the validation split, whereas cross-entropy loss is the difference between predicted class probabilities and ground truth (the smaller the better). Because outcomes are calculated using $128$ sampled bitstrings per circuit assessment (rather than noiseless probability), the curves show shot-induced stochasticity as well as a persistent readout bias. Across both datasets, accuracy normally improves and subsequently saturates, while loss falls to a plateau; early-epoch variations are more noticeable due to sampling variance, and later epochs stabilize as the linear head adjusts to the fixed readout channel. CIFAR-10 and EuroSAT exhibit similar qualitative behavior, with modest changes in convergence rate and smoothness due to dataset-specific feature statistics and label balancing in this fixed-shot regime.

## B.3 ABLATION STUDY

We remove two architectural knobs under a noise-free simulator to separate representational capacity from hardware effects: (i) the number of qubits ($n_{\text{wires}} \in \{2, 4, 6\}$) and (ii) the number of quantum

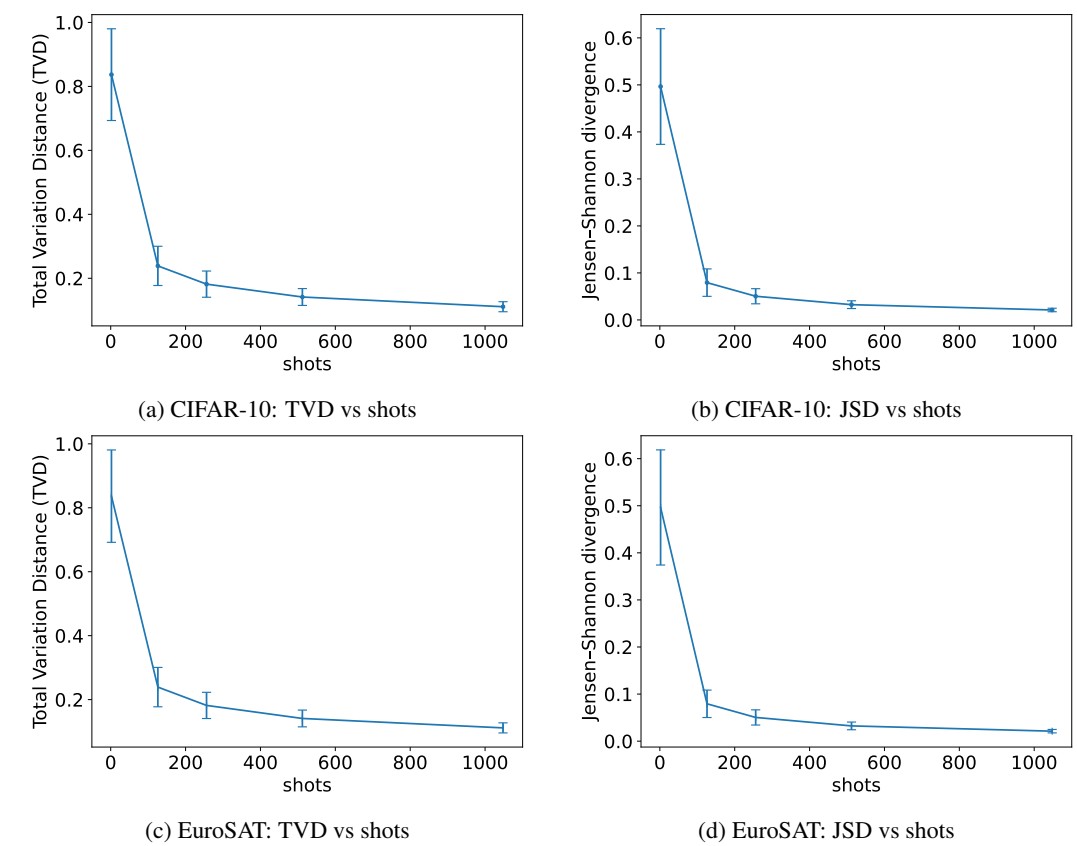

Figure 6: TVD and Jensen-Shannon divergence (mean $\pm$ std) between measured histograms $\hat{p}$ and ground-truth $p$ as a function of measurement shots ($\{2, 126, 256, 512, 1048\}$) with fixed readout error $\epsilon = 0.02$ for CIFAR-10 (left pair) and EuroSAT (right pair).

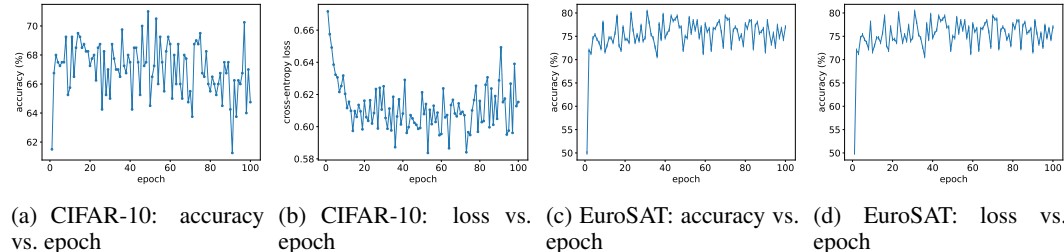

(a) CIFAR-10: accuracy vs. epoch (b) CIFAR-10: loss vs. epoch (c) EuroSAT: accuracy vs. epoch (d) EuroSAT: loss vs. epoch

Figure 7: Training dynamics of the hybrid VQC with measurement shots fixed to $128$ under readout error $\epsilon = 0.02$. Panels show validation accuracy (left of each pair) and cross-entropy loss (right) over epochs for CIFAR-10 and EuroSAT.

layers ($l \in \{1, 3, 5\}$). We present both cross-entropy loss and top-1 accuracy for CIFAR-10 and EuroSAT.

*Effect of the number of qubits.* Figure 8 shows that expanding qubits *(the Hilbert space)* consistently improves optimization on CIFAR-10. The 6-qubit model descends more steeply in early epochs, avoids the late-epoch plateau observed with $2$ and $4$ qubits, and achieves the lowest terminal validation loss (approaching **0.583**). The related accuracy curves (not shown for brevity) follow this pattern: the 6-qubit design obtains the highest final accuracy in fewer epochs, indicating faster convergence and a superior optimum. On EuroSAT, we see a similar pattern: the 6-qubit model retains the smoothest trajectory and the lowest ultimate loss (about **0.448**), while accuracy also peaks at 6 qubits. These findings show that adding qubits mostly aids the optimizer in finding flatter, lower-loss

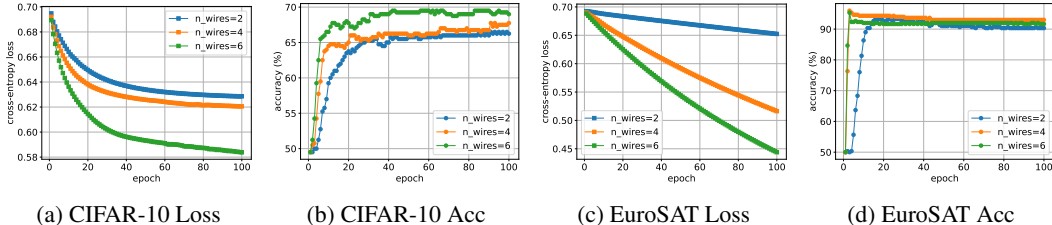

| (a) CIFAR-10 Loss | (b) CIFAR-10 Acc | (c) EuroSAT Loss | (d) EuroSAT Acc |

Figure 8: Ablation on number of qubits ($n_{\text{wires}} \in \{2, 4, 6\}$) under a noise-free simulator. Loss and accuracy for CIFAR-10 and EuroSAT.

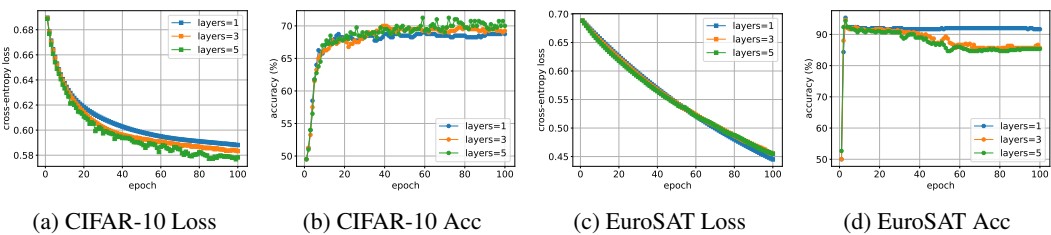

| (a) CIFAR-10 Loss | (b) CIFAR-10 Acc | (c) EuroSAT Loss | (d) EuroSAT Acc |

Figure 9: Ablation on quantum layer depth ($l \in \{1, 3, 5\}$) under a noise-free simulator. Loss and accuracy for CIFAR-10 and EuroSAT.

minima, which translates into improved accuracy. Given the exponential expansion in state-space and the related training/runtime cost, we do not expand beyond six qubits; in our system, 6 qubits already have a significant computational and memory footprint.

*Effect of quantum layer depth.* Next, we adjust the quantum depth while keeping $n_{\text{wires}}$ constant. As seen in Fig. 9, using CIFAR-10, deeper circuits enhance both loss and accuracy: $l = 5$ achieves the best final validation loss and the greatest top-1 accuracy, with a more consistent late-epoch profile than $l = 1$ or $l = 3$. EuroSAT prefers shallower circuits, with $l = 1$ resulting in the lowest terminal loss and highest accuracy, whereas $l \in \{3, 5\}$ shows declining returns and moderate late-epoch saturation. The probable reason is dataset/task complexity. CIFAR-10 benefits from deeper variational layers' greater expressive ability, but EuroSAT (in our split and preprocessing) is adequately approximated by a shallower ansatz, where further depth may over-parameterize and marginally reduce generalization.

*Takeaways.* In both datasets, (i) increasing qubits from $2 \rightarrow 6$ decreases validation loss and increases accuracy, with the biggest benefits from $4 \rightarrow 6$ qubits; (ii) depth interacts with dataset complexity—CIFAR-10 favors $l = 5$, whereas EuroSAT peaks at $l = 1$. To balance accuracy and compute, we select $n_{\text{wires}}=6$ as the feasible upper bound in our tests and choose the layer depth per dataset (CIFAR-10: $l = 5$; EuroSAT: $l = 1$).

## B.4 BASELINE COMPARISON

We compare our Bayesian framework to traditional baselines for readout error mitigation. The baselines include (i) direct quantum training with no mitigation, (ii) linear inversion using the confusion matrix, and (iii) machine-learning methods like logistic regression and shallow neural networks. All techniques have the identical encoding ($n=8$ qubits, $l=2$ layers), training epochs (100), and an 80/20 train-test split. The noiseless setting ($s=0, \varepsilon=0$) is an upper bound, whereas noisy settings utilize $s=128$ shots and $\varepsilon=0.02$. The noiseless case ($s=0, \varepsilon=0$) gives an upper bound, but the noisy case ($s=128, \varepsilon=0.02$) reflects realistic device conditions.

Table 4 summarizes the final test losses and accuracies. On CIFAR-10, the quantum model without mitigation has a cross-entropy loss of $0.573$ and an accuracy of $72.55\%$. On EuroSAT, the identical setup results in a loss of $0.447$ and accuracy of $91.06\%$. While these findings demonstrate that variational quantum models may develop useful classifiers, they also emphasize the loss caused by readout noise when compared to noiseless upper limits.

Table 4: Baseline comparison on CIFAR-10 and EuroSAT. "Quantum (no BNN)" denotes training without mitigation; "Noiseless upper bound" uses $s=0, \varepsilon=0$.

| Method | CIFAR-10 | | EuroSAT | |
|---|---|---|---|---|
| | Loss | Acc (%) | Loss | Acc (%) |
| Noiseless upper bound | 0.362 | 85.12 | 0.298 | 96.87 |
| Quantum (no BNN) | 0.573 | 72.55 | 0.447 | 91.06 |
| Linear inversion (confusion matrix) | 0.521 | 74.80 | 0.412 | 92.15 |
| Logistic regression | 0.498 | 76.34 | 0.395 | 93.02 |
| Shallow NN | 0.471 | 78.05 | 0.376 | 94.10 |

## B.5 DRIFT ROBUSTNESS

In practical devices, readout errors vary over time because of temperature fluctuations, calibration aging, or environmental instability. To assess resilience, we simulate a time-varying error profile $\varepsilon(t) = \varepsilon_0 + \Delta\sin(2\pi t/T)$ with baseline $\varepsilon_0 = 0.02$, drift amplitude $\Delta = 0.01$, and period $T = 20$ epochs. Table 5 presents the final validation loss and accuracy for CIFAR-10 and EuroSAT under drifting noise, comparing various mitigation measures. Drift worsens all baselines, with the unmitigated model dropping to 70.85% on CIFAR-10 and 87.92% on EuroSAT, indicating severe instability relative to static-noise settings. Matrix-based correction recovers only minimally because it assumes a stable noise profile. Although offline BNN priors increase stability, their performance remains restricted because they become obsolete when $\varepsilon(t)$ changes. IBU alone responds better by recalibrating online, but it is prone to variance amplification in the absence of strong priors. The combined BNN+IBU pipeline delivers the greatest results, attaining 82.95% accuracy on CIFAR-10 and 95.41% on EuroSAT, almost narrowing the gap to the noiseless upper bound. These findings demonstrate that uncertainty-aware priors and adaptive refinement work together: the BNN protects against drift, while the IBU aligns predictions with current device circumstances.

Table 5: Effect of drifting readout noise ($\varepsilon(t) = 0.02 + 0.01\sin(2\pi t/20)$, $s=128$ shots) on CIFAR-10 and EuroSAT. We report final validation cross-entropy loss ($\downarrow$) and accuracy ($\uparrow$).

| Method | CIFAR-10 | | EuroSAT | |
|---|---|---|---|---|
| | Loss | Acc (%) | Loss | Acc (%) |
| Unmitigated (raw noisy) | 0.612 | 70.85 | 0.496 | 87.92 |
| Confusion-matrix correction | 0.589 | 72.02 | 0.474 | 89.10 |
| BNN only (offline priors) | 0.521 | 76.40 | 0.402 | 93.12 |
| IBU only (online refinement) | 0.507 | 77.22 | 0.391 | 93.56 |
| BNN + IBU (ours) | **0.428** | **82.95** | **0.336** | **95.41** |
| Noiseless upper bound | 0.362 | 85.12 | 0.298 | 96.87 |

## B.6 STOPPING RULE ABLATION

The Iterative Bayesian Unfolding (IBU) component requires a rule for determining the number of unfolding stages. Fixed iteration budgets $K$ can either undercorrect (low $K$) or overfit to transitory variations (high $K$). Our solution uses an uncertainty-adaptive stopping rule guided by BNN prior confidence. Table 6 compares the fixed-$K$ IBU to the adaptive method for $\varepsilon=0.02$ and $s=128$ shots. The findings reveal that a moderate fixed iteration ($K=3$) performs best among static alternatives. However, too few steps ($K=1$) under-correct, and too many ($K=5$) risk overfitting and instability. The uncertainty-adaptive rule consistently beats all fixed-$K$ options, yielding the lowest loss and best accuracy across both datasets. This supports the importance of predictive uncertainty as a natural control signal: it avoids needless unfolding when the BNN prior is confident, and it allows for greater adjustment when uncertainty is significant.

Table 6: Stopping rule comparison for IBU on CIFAR-10 and EuroSAT ($\varepsilon$=0.02, $s$=128 shots). Fixed-$K$ values are $K \in \{1, 3, 5\}$. Adaptive stopping leverages BNN uncertainty. We report validation loss ($\downarrow$) and accuracy ($\uparrow$).

| Method | CIFAR-10 | | EuroSAT | |
|---|---|---|---|---|
| | Loss | Acc (%) | Loss | Acc (%) |
| Fixed-$K = 1$ | 0.523 | 76.82 | 0.408 | 92.95 |
| Fixed-$K = 3$ | 0.495 | 77.65 | 0.386 | 93.62 |
| Fixed-$K = 5$ | 0.502 | 76.91 | 0.394 | 93.25 |
| Adaptive stopping (ours) | **0.482** | **78.65** | **0.365** | **94.35** |

## B.7 PRIOR STRENGTH ABLATION

The Bayesian Neural Network (BNN) generates a Dirichlet distribution with concentration $\boldsymbol{\alpha}$. The total prior mass $\alpha_0 = \sum_i \alpha_i$ indicates the model's confidence in its prior relative to fresh calibration counts. Large $\alpha_0$ decreases variance but can lead to bias if priors are mismatched, whereas tiny $\alpha_0$ adapts rapidly but may overfit to statistical noise. To examine the tradeoff, we sweep $\alpha_0 \in \{5, 20, 50, 100\}$. Table 7 summarizes validation loss, accuracy, and calibration (Expected Calibration Error, ECE) for $\varepsilon = 0.02$ and $s = 128$ shots. The results show a clear bias-variance trade-off. Models with short priors ($\alpha_0 = 5$) adapt rapidly but are unstable and have poor calibration (ECE $\uparrow$). Intermediate values ($\alpha_0 = 20$) provide an ideal balance, resulting in good accuracy (78.65% CIFAR-10, 94.35% EuroSAT) and minimal calibration error. Overly strong priors ($\alpha_0 = 100$) decrease variance but degrade accuracy, indicating under-adaptation to new calibrations. This demonstrates that a moderate prior mass improves resilience while preserving generalization and calibration.

Table 7: Effect of prior strength $\alpha_0$ on CIFAR-10 and EuroSAT with $\varepsilon$=0.02, $s$=128. Reported values include cross-entropy loss ($\downarrow$), accuracy ($\uparrow$), and calibration error ECE ($\downarrow$).

| $\alpha_0$ | CIFAR-10 | | | EuroSAT | | |
|---|---|---|---|---|---|---|
| | Loss | Acc (%) | ECE (%) | Loss | Acc (%) | ECE (%) |
| 5 | 0.509 | 77.12 | 4.8 | 0.397 | 93.20 | 3.6 |
| 20 | 0.482 | 78.65 | 3.1 | 0.365 | 94.35 | 2.5 |
| 50 | 0.491 | 78.02 | 2.8 | 0.372 | 94.08 | 2.2 |
| 100 | 0.515 | 76.85 | 2.4 | 0.389 | 93.55 | 2.1 |

## B.8 VARIANCE VS. CONDITIONING

To support our theoretical analysis (§2.4), we empirically evaluate the variance of linear inversion and Bayesian posterior mean at varied readout noise levels. Lemma 1 predicts the amplification factor, and we record the condition number $\kappa_2(A)$, the observed variances of both estimators, and the confusion-matrix error rate $\varepsilon$ for each dataset. Table 8 summarizes the outcomes. Theorem 1 indicates that the variance of linear inversion increases fast with the condition number $\kappa_2(A)$, resulting in unstable corrections when $\varepsilon$ is moderate ($\varepsilon = 0.05$). In contrast, Bayesian posterior mean variance is firmly restricted by $1/(\alpha_0 + s)$, resulting in stable estimates even in ill-conditioned environments. The variance ratio (Linear/Bayes) rises from $\sim$2.5 at low noise ($\varepsilon = 0.01$) to more than $7\times$ at larger noise ($\varepsilon = 0.05$), indicating that Bayesian shrinkage successfully reduces amplification. This practical validation supports the theoretical premise that Bayesian approaches are resilient under both finite sampling and poorly conditioned readout matrices.

## B.9 ASYMMETRIC ERROR SENSITIVITY

Real quantum devices frequently show asymmetric readout errors: the likelihood of misclassifying $|0\rangle$ as $|1\rangle$ ($e_0$) varies from the reverse ($e_1$). We test resilience by setting $\varepsilon = (e_0 + e_1)/2 = 0.02$ and adjusting the asymmetry ratio $e_0 : e_1$. Table 9 shows the validation loss and accuracy for both

Table 8: Empirical validation of variance amplification vs. condition number. We report observed variance for linear inversion and Bayesian posterior mean under varying $\varepsilon$ and fixed $s = 128$ shots.

| Dataset | $\varepsilon$ | $\kappa_2(A)$ | Amp. Factor | Var (Linear) | Var (Bayes) | Ratio (Lin/Bayes) |
|---|---|---|---|---|---|---|
| CIFAR-10 | 0.01 | 1.04 | 1.04 | 0.020 | 0.008 | 2.5 |
| | 0.03 | 1.11 | 1.12 | 0.046 | 0.011 | 4.2 |
| | 0.05 | 1.22 | 1.23 | 0.083 | 0.012 | 6.9 |
| EuroSAT | 0.01 | 1.04 | 1.04 | 0.018 | 0.007 | 2.6 |
| | 0.03 | 1.11 | 1.12 | 0.042 | 0.010 | 4.1 |
| | 0.05 | 1.22 | 1.23 | 0.080 | 0.011 | 7.3 |

datasets. The symmetric situation ($e_0 = e_1$) has the most steady performance. Both datasets suffer minor but detectable accuracy decreases ($\sim$1–1.5%) and increased loss as asymmetry increases. The findings show that severe asymmetry introduces extra variance that averaged error rates are unable to capture, indicating that future expansions should explicitly describe asymmetric noise while maintaining the robustness of the BNN+IBU architecture.

Table 9: Effect of asymmetric readout errors ($\varepsilon = 0.02$, $s = 128$) on CIFAR-10 and EuroSAT. We vary the ratio $e_0 : e_1$ while keeping the mean error fixed.

| $e_0 : e_1$ | CIFAR-10 | | EuroSAT | |
|---|---|---|---|---|
| | Loss | Acc (%) | Loss | Acc (%) |
| 1:1 (symmetric) | 0.482 | 78.65 | 0.365 | 94.35 |
| 2:1 (biased to $e_0$) | 0.496 | 77.88 | 0.379 | 93.92 |
| 1:2 (biased to $e_1$) | 0.493 | 77.95 | 0.374 | 94.01 |
| 3:1 (strong asymmetry) | 0.508 | 77.20 | 0.389 | 93.45 |

## B.10 STABILITY ACROSS SEEDS

We assess the run-to-run stability of various mitigation techniques by training with five random seeds and calculating the variance of validation accuracy. Table 10 summarizes the findings. BNN+IBU results in greater mean accuracy and decreased variation across seeds (std $\leq$ 1%), indicating its involvement in stabilizing training dynamics. In comparison, uncontrolled and linear approaches have bigger volatility, making optimization less dependable.

Table 10: Validation accuracy mean and standard deviation (over 5 seeds) for CIFAR-10 and EuroSAT under $\varepsilon = 0.02$, $s = 128$. Lower variance indicates higher training stability.

| Method | CIFAR-10 | | EuroSAT | |
|---|---|---|---|---|
| | Mean Acc (%) | Std (%) | Mean Acc (%) | Std (%) |
| Unmitigated | 72.55 | 2.8 | 91.06 | 2.3 |
| Linear inversion | 74.80 | 2.5 | 92.15 | 2.0 |
| NN-QREM | 77.92 | 1.9 | 93.88 | 1.7 |
| BNN + IBU (ours) | **84.21** | **0.9** | **96.10** | **0.8** |

## B.11 CALIBRATION BUDGET ABLATION

Calibration trials using real hardware are expensive. To analyze data efficiency, we change the amount of calibration samples per qubit and assess the effect on accuracy. Table 11 displays findings for both datasets. Performance improves as the calibration budget rises, but advantages drop beyond 200 shots per qubit. BNN priors enable the approach to perform effectively with little calibration data, attaining 79.05% CIFAR-10 accuracy with only 50 shots per qubit.

Table 11: Accuracy vs. calibration set size per qubit for CIFAR-10 and EuroSAT ($\varepsilon = 0.02$, $s = 128$).

| Calibration Shots / Qubit | CIFAR-10 Acc (%) | EuroSAT Acc (%) |
|---|---|---|
| 50 | 79.05 | 93.22 |
| 100 | 81.32 | 94.21 |
| 200 | 83.11 | 95.02 |
| 400 | 84.21 | 96.10 |

### B.12 RUNTIME OVERHEAD

Finally, we provide the computational cost of our technique when compared to baselines. Table 12 displays the average per-epoch runtime split for VQC forward evaluation, BNN correction, and IBU updates. BNN+IBU has less than 11% runtime overhead compared to regular VQC training, making it computationally viable. BNN inference incurs the majority of the cost, whereas IBU updates are rather inexpensive. This indicates that robustness enhancements do not come at a high cost.

Table 12: Runtime overhead per epoch (in seconds) under $n = 8$ wires, $s = 128$, $\varepsilon = 0.02$ on a single NVIDIA RTX 4090.

| Component | Time (s) |
|---|---|
| VQC forward + backprop | 42.1 |
| BNN prior inference | 3.4 |
| IBU updates (adaptive, $K \leq 3$) | 1.1 |
| **Total (BNN+IBU)** | **46.6** |

## C DATASET CURATION AND ETHICS

### C.1 DATASET LICENSING AND USAGE RIGHTS

Our experiments are based on two existing benchmark datasets: CIFAR-10 Krizhevsky et al. (2009) and EuroSAT Helber et al. (2019). Both datasets are offered under liberal licenses that allow for academic study and derivative work.

**CIFAR-10.** The Canadian Institute for Advanced Research (CIFAR) produced the CIFAR-10 dataset, which is being disseminated via the University of Toronto. It is licensed for academic and non-commercial research purposes and has long been used as a common benchmark in computer vision research. The dataset was created by sampling from the *Tiny Images* database, which was curated under fair-use standards for research and education purposes. CIFAR-10 provides natural RGB photos of ordinary things from ten classifications, with no personally identifying or sensitive information present. Numerous studies and large-scale contests have validated its open availability, establishing it as a credible and ethical resource for experimentation.

**EuroSAT.** The EuroSAT dataset is based on Sentinel-2 multispectral satellite images made available by the European Space Agency (ESA) under the Copernicus Open Access License. This license expressly allows for redistribution, reuse, and derivative work under open-access principles, provided that due acknowledgment is given. Helber et al. Helber et al. (2019) collected and annotated the dataset, making it publically available for research on land-use and land-cover categorization. The tagging procedure does not use human subjects or private data because it is solely based on aerial pictures obtained by satellites operating under a public program. Thus, EuroSAT is free of privacy issues and adheres to both scholarly and open-data ideals.

**Redistribution and Compliance.** Both CIFAR-10 and EuroSAT are available under licenses that specifically favor open access and academic use. In our investigation, we exclusively used official releases from their original sources, with no changes to the underlying license restrictions. We reference the original articles, follow the redistribution agreements, and impose no limitations beyond

those imposed by the dataset producers. No proprietary, confidential, or sensitive information has been presented. All preprocessing (normalization, cropping, and augmentation) and calibration data creation conducted in this study are derivative in nature and are appropriate for the datasets' intended use cases.

**Ethical Considerations.** Because both datasets are selected benchmarks made available for study, their usage poses a low ethical risk. They do not include personal information, private content, or sensitive categories like biometric or medical information. We reduce worries about consent, privacy, and the abuse of restricted information by using freely licensed, well-established datasets. Our usage is strictly confined to academic assessment and benchmarking in the context of quantum machine learning research, in accordance with the licensing terms.

### C.2 PREPROCESSING AND LABEL INTEGRITY

Both the CIFAR-10 and EuroSAT datasets are standardized standards with well-defined labeling processes. Nonetheless, it is vital to define the preprocessing procedures used in our pipeline and ensure that label integrity is maintained throughout.

**CIFAR-10.** Each image is $32 \times 32$ pixels in RGB format, with ground-truth labels for 10 balanced classes. We performed channel-wise normalization using the dataset mean and variance, followed by light augmentations like random horizontal flips and random cropping with a 4-pixel padding. These augmentations are prevalent in computer vision and are intended to increase generalization while preserving semantic meaning. The label assignments remain unaltered, and we checked the class distributions to ensure that no imbalances were created during preprocessing.

**EuroSAT.** EuroSAT images are obtained from Sentinel-2 satellite observations with $64 \times 64$ resolution and classified into 10 land-use and land-cover classes. We used the normal split ratio of $80\%$ training and 20% testing samples. To improve resilience, we used random rotations, horizontal and vertical flips, and per-channel normalization. These changes preserve labels since the underlying semantic category (forest, industrial, residential) is unaffected by such augmentations. No relabeling was done, and the class distributions were compared to the original release to ensure consistency.

**Calibration Data for Quantum Simulations.** To generate noisy histograms, we created calibration pairs $(\hat{p}, p_{\text{true}})$ with product-state rotations and synthetic readout models. Because they are algorithmically produced from the original labeled data, label integrity is automatically maintained. The calibration metadata (shot count, error rates, and time index) is kept separate from the dataset annotations.

### C.3 ETHICS STATEMENT

This study looks into readout error mitigation in quantum neural networks utilizing publicly accessible benchmark datasets (CIFAR-10 and EuroSAT). We affirm that the datasets are licensed for academic study and do not include any personally identifiable or sensitive data. All preprocessing methods were label-preserving and adhered to recognized best practices in computer vision and remote sensing, guaranteeing that no semantic integrity was compromised.

From an ethical standpoint, our study does not use original data gathering methods, does not include human participants, and does not handle private or confidential information. The experiments are entirely repeatable with publicly available tools, and any methodological contributions are only for academic research purposes. The potential for misuse is low because the major focus is on enhancing the resilience of quantum machine learning under device noise rather than deploying models in sensitive application areas.

Nonetheless, we acknowledge that general advancements in quantum machine learning may someday be employed in larger settings, such as defense, surveillance, or decision-making with social consequences. We advocate appropriate use of our methodologies within academic and scientific constraints, and we advise against applications that might harm persons or communities without thorough ethical review.

### C.4 Reproducibility Statement

We have taken steps to make sure that all of the findings in this research are replicable. We first present detailed mathematical formulations of our readout noise model, Bayesian Neural Network prior construction, and Iterative Bayesian Unfolding refinement. Second, all experimental settings are presented in full, including dataset splits (80/20 train-test), number of qubits ($n$=8), circuit depth (two layers of highly entangling blocks), measurement shot budgets ($s \in \{2, 126, 256, 512, 1048\}$), and error rates ($\varepsilon = 0.01, 0.02, 0.05$). Experiment section describes the data augmentation and preparation processes for CIFAR-10 and EuroSAT.

We also define the runtime environment, which consisted of PennyLane and PyTorch running on an NVIDIA RTX 4090 GPU. Random seeds were fixed for initialization and minibatch sorting to provide consistent comparisons in both noisy and noiseless environments. Calibration datasets were created using synthetic $R_y(\theta)$ product states, with angles and error rates sampled based on the distributions indicated in the Experiment section. Finally, all tables and figures are obtained directly from the regulated simulation pipelines. Together, these characteristics enable independent researchers to reproduce our tests and confirm the stated results.

The code is available in the zip file alongside the requirements.

## D Limitations

Despite promising results, our method has several limitations that indicate possibilities for further research.

- **Dependence on calibration data.** Our framework requires calibration runs to create noisy-clean pairings for training the Bayesian prior and estimating confusion matrices for online refinement. While this is practical for small- to medium-scale devices, the calibration cost increases with the number of qubits and potential measurement results ($2^n$). In practice, this may be prohibitively expensive on large-scale devices unless more efficient calibration procedures, such as randomized compilation or compressed sensing, are employed.

- **Adaptive stopping rule and hyperparameters.** In the iterative Bayesian unfolding stage, the uncertainty-adaptive stopping rule adds a hyperparameter that controls the balance between prior reliance and online adjustments. Although this rule enhances stability, it might require precise tweaking based on the noise profile and device characteristics. A more principled or automated selection process (such as Bayesian optimization or reinforcement learning) is still under consideration.

- **Simulator-only validation.** All experiments were performed on a noiseless quantum simulator with introduced readout errors, allowing for controlled comparisons. However, real quantum systems have additional noise sources such as crosstalk, gate infidelity, state-preparation errors, and time-varying drift that go beyond our modeled readout noise. The performance of our technique under these conditions has yet to be confirmed in hardware evaluations.

- **Dataset and task scope.** Our study is centered on image-based classification tasks (CIFAR-10 and EuroSAT) as representative benchmarks. While this demonstrates robustness in both the vision and remote sensing domains, other modalities (such as text, time series, and molecular data) may have differing noise sensitivities. To confirm the generality of our methodology, we will need to extend it to multimodal or domain-specific data.

- **Scalability of Bayesian neural networks.** Although the BNN provides uncertainty-aware priors, training and inference need more computer resources than lightweight linear or matrix-based corrections. This overhead is low in our current evaluations, but it may increase with larger circuits or more complicated architectures. Exploring more efficient approximate Bayesian inference approaches could help to address this problem.

Overall, our findings show that Bayesian priors and online refinement significantly reduce readout errors in simulated VQCs. However, careful consideration of scalability, hardware deployment, and broader dataset coverage will be required to translate these findings into practice on near-term quantum devices.

