# OpenReview forum: "Readout Noise Mitigation with Bayesian Methods for Quantum Neural Networks"
_ICLR.cc/2026/Conference — ICLR 2026 Conference Withdrawn Submission_

### Official Review · Reviewer_prMY · 2025-10-17

**Soundness:** 2
**Presentation:** 1
**Contribution:** 3
**Rating:** 2
**Confidence:** 4

**Summary:**

This paper proposes a two-stage Bayesian framework to mitigate readout noise in variational quantum circuits (VQCs) and quantum neural networks (QNNs), a key source of performance degradation in near-term quantum hardware. The paper tackles the problem of readout noise—i.e., misclassifications in qubit measurements—which leads to biased gradients, reduced accuracy, and destabilized training. Existing mitigation methods like confusion-matrix correction and ML post-processing struggle with calibration drift, limited data, and uncertainty quantification. The paper combines two key contributions, i.e., offline Bayesian Neural Network (BNN) and online Iterative Bayesian Unfolding (IBU). The authors aim to demonstrate that combining offline Bayesian priors with online adaptive updating provides a scalable and robust strategy for noise mitigation in dynamic, non-stationary environments.

**Strengths:**

- The problem addressed in the paper is rather crucial and relevant in the context of quantum computing.
- The idea of addressing readout measurements error with Bayesian learning techniques is sound and promising.
- The paper aims at being sound both on the theoretical and on the experimental level (theorems and proofs are provided as well as an extensive set of experiments).
- Limitations are clearly mentioned and explained.

**Weaknesses:**

- The presentation is often lacking clarity. The writing style is at times hard to follow.
- Many concepts are repeated any times (copy pasting exact same sentence several times). While it is good to reiterate some concepts I think it may be confusing if overdone.
- Rigour is often lacking. Same variables are used twice for different observables within two subsequent section thus leading to crucial confusion if one is not aware of it.
- A lot of necessary context about quantum computing is missing. For example, the authors talk about shots before even defining what is meant with that. They also talk about rotation and angles, without ever mentioning what these angles are. They talk about variational quantum circuits but no context is given (what they are, what they do.). For reference the authors can look into [this paper](https://proceedings.neurips.cc/paper_files/paper/2023/file/3adb85a348a18cdd74ce99fbbab20301-Paper-Conference.pdf) from NeurIPS 2023. The authors also discuss bayesian optimisation in the context of variational quantum algorithms but they provide a very detail introduction to the topic for making it accessible to a broader audience. The gap between machine learning and quantum computing is in my opinion currently too big in the present paper to make it accessible.

**Questions:**

### Additional concerns: wiriting

- The text is at times very unclear, general and possibly even confusing. For example, in line 41-42 the authors state:
"[...] resulting in a nonzero possibility of an error."

this sentence is highly confusing. First, possibility is a bit of a weird word to use in this context (I'd rather go for probability as it is far more appropriate). Second, "error" is a bit of a general term here and in order to allow the reader to fully grasp the goal of the paper since the very beginning I would recommend to be a bit more specific here.

There are many other examples where the writing require a substantial round of revision but I'll omit all those for brevity here. I can only encourage the authors to have a thorough round of proof-reading of their paper.

- Some words often sound strange in the context they are in. For example, "dependability" in line 40.

### Additional concerns: refs
I believe that there's a substantial body of literature about bayesian learning for mitigating readout noise that the authors may have overlooked. This includes but is not limited to:
- [Nicoli et al., NeurIPS (2023)](https://scholar.google.com/citations?view_op=view_citation&hl=it&user=0GzYud8AAAAJ&citation_for_view=0GzYud8AAAAJ:KlAtU1dfN6UC)
- [Anders et al., ICML (2024)](https://proceedings.mlr.press/v235/anders24a.html)
- [Tamiya et al., npj quantum](https://www.nature.com/articles/s41534-022-00592-6)
- [Nicoli et al., arXiv preprint arXiv:2501.17689 (2025)](https://arxiv.org/pdf/2501.17689?)
- [Pedrielli et al. arXiv preprint arXiv:2502.02625 (2025)](https://arxiv.org/pdf/2502.02625)

I'd recommend the authors to perform a thorough round of literature search to spot any other related work and make sure to have an appropriate related work section in the paper.

### Additional concerns: questions

- I do not quite get what the authors mean with "Errors from defective gates, crosstalk, decoherence, and calibration drift all de-
grade processing; however, readout noise is more vital Chen et al. (2023)." Seems as if they suggest readout noise is more important than hardware noise but I am not sure whether this is really the case.

- Line 83: shots is mentioned for the first time and has not been defined anywhere.
- Often enough the authors use the word "ideal" where is in my opinion more appropriate to use the word "true" as this often refers to the true target distribution (see e.g., last line in page 2). Please clarify if I am misunderstanding something and there's a reason for using the word ideal. In fact there are multiple other occurrences in the paper for similar or same examples.
- Line 115: when the authors say "correlated" do they mean "entangled"?
- Sometimes author refer to "accurate" and "correct" probability distribution. What do they mean with this?
- The mathematical section around equation (3) does not appear to be very rigorous in both math and text. Same hold for some later section of the paper where a lot of mathematical derivations are displayed.
- Same letter $\theta$ is used to indicate both parameters of a VQC (not explicitly mentioned though) and also the weights of the BNN. This is very confusing especially for people not familiar with the topic of quantum computing.
- What do the authors mean with computation foundation on line 168?
- In equation 9 I do not understand why we compute the KL divergence between the variational posterior and the prior. Shouldn't one want to compute KL between variational posterior and target posterior, i.e., true distribution?
- I find section 2.4 a bit cluttered and at times hard to follow. I'd think it'd make it more understandable if the authors would break have this section less condensed and maybe focus on few key aspects and relegate the rest to the appendix.
- I find the formatting for text in lines 310 and 313 a bit strange and perhaps also the text to be not necessary as this refers to variable name that may only be relevant and interesting at an implementation level.

**Details Of Ethics Concerns:**

No ethics concern.

---

### Official Review · Reviewer_Da5m · 2025-10-29

**Soundness:** 2
**Presentation:** 3
**Contribution:** 2
**Rating:** 2
**Confidence:** 3

**Summary:**

This paper proposes a two-stage Bayesian approach for mitigating readout noise in variational quantum circuits and quantum neural networks. The method combines two part: an offline Bayesian Neural Network that learns uncertainty-aware priors from historical calibration data, and an online Iterative Bayesian Unfolding (IBU) step with uncertainty-adaptive stopping rules. The experiment results on CIFAR-10 and EuroSAT datasets, demonstrating improvements over existing mitigation techniques.

**Strengths:**

1.The comparison between Bayesian and linear inversion methods (Theorem 1) provides valuable insights into variance-bias tradeoffs.
2.The BNN framework naturally provides uncertainty estimates, enabling adaptive iteration control in the IBU phase.

**Weaknesses:**

1.No convergence proof or convergence rate analysis for IBU iterations is provided. Uncertainty-adaptive stopping rules lack theoretical guarantees and may lead to premature stopping or excessive iterations.
2. The use of a single-layer Ry rotation gate to generate calibration data is too simplistic and does not represent the output distribution of a real VQC. There is no analysis of the generalization ability of the method to different types of quantum circuits (e.g., parameterized quantum circuits vs. fixed circuits).
3. The "locally well-specified prior" assumption in Assumption 2 lacks a rigorous definition.

**Questions:**

1. What is the specific strategy for adaptive shutdown of online IBUs? How are thresholds based on α_0 or ECE/JSD set and stabilized?
2. How does BNN+IBU utilize/estimate the correlation terms compared to the matrix baseline under correlation readout? Can you provide an online estimation and uncertainty propagation of the correlation calibration matrix A?
3. If the calibration data for training BNNs comes only from direct product states, how can we ensure generalization to entangled distributions?
4. Under what conditions is the prior "locally well-specified"? How can the assumption that bias = o(1/(α₀+s)) be verified?

---

### Official Review · Reviewer_65jH · 2025-10-30

**Soundness:** 2
**Presentation:** 2
**Contribution:** 2
**Rating:** 6
**Confidence:** 1

**Summary:**

This paper proposes a two-stage Bayesian readout noise mitigation framework for addressing readout noise and drift in VQC/QNN training: the offline stage uses BNN to map noisy histograms to corrected distributions and provides uncertainty estimates

**Strengths:**

1. Clear problem motivation + structured solution: Intuitively explains the impact of readout noise on optimization and gradients, proposing a combined pipeline of “offline BNN prior + online IBU refinement” with logically consistent reasoning

**Weaknesses:**

1. Lack of real hardware verification (core): All results were achieved on simulators
2. The Gap Between Theory and Practice: Theoretical conditions (such as the MSE comparison theorem) are based on assumptions of locally well-defined priors and known/stable A. However, in actual implementations, A may involve non-independent tensor products and contain correlated errors, while BNN priors may deviate from the true p
3. Task Selection and Generalizability of Metrics: CIFAR-10 and EuroSAT are classic image classification datasets, encoded into 8-bit, 2-layer entangled structures via angle coding. Whether the gains observed in these small-scale settings can be extrapolated to deeper circuits, more bits, or more complex encodings remains uncertain.

**Questions:**

check the weakness

By the way, I'm not in the doamin of quantum

PC and SPC can ignore my comment if you think my comments are not correct

And I have to say the code is difficult to navigate, the repository lacks clear structure and step-by-step guidance (README, setup, and end-to-end run scripts). As a result, I couldn’t find a reliable path to reproduce the main results

---

### Official Review · Reviewer_zsaG · 2025-11-01

**Soundness:** 2
**Presentation:** 2
**Contribution:** 2
**Rating:** 2
**Confidence:** 4

**Summary:**

This paper tackles measurement readout noise in VQCs/QNNs, which distorts output distributions and degrades training, especially under finite shots and time-varying (non-stationary) noise. The authors propose a two-stage Bayesian pipeline: (i) an offline Bayesian neural network (BNN) that maps noisy histograms to a Dirichlet prior over the true outcome distribution with calibrated uncertainty (including a shot-aware penalty that scales the Dirichlet mass), and (ii) an online Iterative Bayesian Unfolding (IBU) step that refines the correction using fresh calibration counts with uncertainty-adaptive early stopping. They provide a theory showing when the Bayesian estimator attains lower MSE than linear inversion (variance of the latter grows with the readout matrix condition number), and validate on CIFAR-10 and EuroSAT with controlled readout error and shot budgets.

**Strengths:**

1. The motivation is well clarified and clearly formulated.
2. Using a Dirichlet prior from a BNN is natural for simplex-valued outputs.
3. Pseudocode for calibration generation, BNN training (Algorithm 2), drift-aware inference, and the full BNN+IBU pipeline enhances readability and implementability.

**Weaknesses:**

1. The introduction largely contrasts confusion-matrix correction vs. generic ML approaches. This misses finer distinctions that would better position the contribution and clarify novelty along the axes of uncertainty modeling, drift adaptivity, and scalability.
2. The scope is limited. The method should be applied to other tasks. Evaluating at least one non-classification setting would broaden the impact.
3. In Algo. 2, two losses and one penalty term are introduced, but their sensitivities or robustness are not analyzed.
4. The comparison set is reasonable (inversion, logistic regression, shallow NN, probabilistic models), but the field has also explored more structure-aware or drift-adaptive readout mitigation; engaging those would better stress-test the approach.

**Questions:**

Besides referring to the above weaknesses, please also consider the following questions:
1. In simulation, did you use PennyLane’s noise backend or a custom implementation that samples counts after applying your constructed confusion matrices? If both were tried, how do the outcomes and runtimes compare (with/without the noise backend enabled)?
2. How does the adaptive stopping behave under faster or non-sinusoidal drift?
3. Have you evaluated scenarios with pairwise-correlated readout errors?

---

### Note · Authors · 2025-11-12

I have read and agree with the venue's withdrawal policy on behalf of myself and my co-authors.